# *MYC* and *Twist1* cooperate to drive metastasis by eliciting crosstalk between cancer and innate immunity

Renumathy Dhanasekaran[1†], Virginie Baylot[2,3†], Minsoon Kim[2,3], Sibu Kuruvilla[2,3], David I Bellovin[2,3], Nia Adeniji[2,3], Anand Rajan KD[4], Ian Lai[2,3], Meital Gabay[2,3], Ling Tong[2,3], Maya Krishnan[2,3], Jangho Park[2,3], Theodore Hu[2,3], Mustafa A Barbhuiya[5,6,7,8], Andrew J Gentles[9,10], Kasthuri Kannan[11,12], Phuoc T Tran[5,6,7,8‡], Dean W Felsher[2,3‡*]

[1]Division of Gastroenterology and Hepatology, Stanford University, Stanford, United States; [2]Division of Oncology, Department of Medicine, Stanford University, Stanford, United States; [3]Division of Oncology, Department of Pathology, Stanford University, Stanford, United States; [4]Department of Pathology, University of Iowa Hospitals and Clinics, Iowa City, United States; [5]Radiation Oncology and Molecular Radiation Sciences, Johns Hopkins University School of Medicine, Baltimore, United States; [6]The Sidney Kimmel Comprehensive Cancer Center, Johns Hopkins University School of Medicine, Baltimore, United States; [7]The James Buchanan Brady Urological Institute, Johns Hopkins University School of Medicine, Baltimore, United States; [8]Department of Urology, Johns Hopkins University School of Medicine, Baltimore, United States; [9]Department of Medicine (Biomedical Informatics), Stanford University School of Medicine, Stanford, United States; [10]Department of Biomedical Data Sciences, Stanford University School of Medicine, Stanford, United States; [11]Department of Pathology, NYU Langone Medical Center, New York, United States; [12]Genome Technology Center, NYU Langone Medical Center, New York, United States

*For correspondence:
dfelsher@stanford.edu

†These authors contributed equally to this work
‡These authors also contributed equally to this work

Competing interests: The authors declare that no competing interests exist.

**Abstract** Metastasis is a major cause of cancer mortality. We generated an autochthonous transgenic mouse model whereby conditional expression of *MYC* and *Twist1* enables hepatocellular carcinoma (HCC) to metastasize in >90% of mice. *MYC* and *Twist1* cooperate and their sustained expression is required to elicit a transcriptional program associated with the activation of innate immunity, through secretion of a cytokinome that elicits recruitment and polarization of tumor associated macrophages (TAMs). Systemic treatment with Ccl2 and Il13 induced *MYC*-HCCs to metastasize; whereas, blockade of Ccl2 and Il13 abrogated *MYC/Twist1*-HCC metastasis. Further, in 33 human cancers (n = 9502) *MYC* and *TWIST1* predict poor survival (p=4.3×10$^{-10}$), CCL2/IL13 expression (p<10$^{-109}$) and TAM infiltration (p<10$^{-96}$). Finally, in the plasma of patients with HCC (n = 25) but not cirrhosis (n = 10), CCL2 and IL13 were increased and IL13 predicted invasive tumors. Therefore, *MYC* and *TWIST1* generally appear to cooperate in human cancer to elicit a cytokinome that enables metastasis through crosstalk between cancer and immune microenvironment.

## Introduction

Tumorigenesis is caused by specific oncogenes but tumor progression often involves the acquisition of metastasis (*Chaffer and Weinberg, 2011*). Metastasis occur when tumor cells gain the ability to invade, migrate and colonize distant sites, and this accounts for most of the morbidity and mortality

**eLife digest** Cancer develops when cells in the body gain mutations that allow them to grow and divide rapidly and uncontrollably. As the disease progresses these cancer cells develop the ability to spread around the body. This process of spreading, called metastasis, is responsible for most cancer-related deaths in humans, but no current treatments target it.

Mutations that increase the levels of two proteins known as MYC and TWIST1 in cells cause many human cancers. In healthy adult cells, normal levels of MYC and TWIST1 act as key regulators that switch thousands of genes on or off. TWIST1 is known to control the movement and spread of cells in the embryo. However, it is not known how MYC and TWIST1 work together to promote the metastasis of cancer cells.

To address this question, Dhanasekaran, Baylot et al. used mice to investigate the roles of MYC and TWIST1 in the metastasis of cancer cells. The experiments showed that these two proteins work together to reprogram mouse cancer cells to release signal molecules known as cytokines. These molecules convert immune cells known as macrophages to a tumor-friendly state that allows cancers cells to spread around the body. Inhibiting two cytokines known as CCL2 and IL13 prevented the cancer cells from moving.

Further experiments analyzed tumor samples from around 10,000 human patients with 33 different cancers. This revealed that patients that had higher levels of MYC and TWIST1 proteins in their tumors also had increased levels of CCL2 and IL13, more activated macrophages and were less likely to recover from their cancer.

The findings of Dhanasekaran, Baylot et al. suggest that MYC and TWIST1 may instigate metastasis in many human cancers, and therapies targeting specific cytokines may prevent these cancers from spreading around the body. Furthermore, screening blood for the levels of cytokines may help to identify the cancer patients who would benefit from such therapies.

associated with cancer (*Mehlen and Puisieux, 2006*). Many studies have examined human clinical specimens and/or tumor-derived cell lines to discern mechanisms of metastasis, and identified the role of specific genes (*Ji et al., 2007*; *Sun et al., 2018*; *Zhu et al., 2017*) and also the role of the tumor microenvironment (*Fidler, 2003*; *Kalluri, 2016*; *Kim et al., 2017*; *Whitfield and Soucek, 2012*). However, to date, there are very few mouse models that exhibit spontaneous metastasis, and even fewer in vivo models where the stepwise progression from a non-metastatic to metastatic cancer can be studied. Such a model would provide a tractable approach for studying specific mechanisms of metastasis, particularly the role of the immune microenvironment. Innate immune cells, especially tumor associated macrophages (TAMs), are known to contribute to metastasis through multiple mechanisms including effects on angiogenesis, production of specific cytokines, suppression of the immune system, and induction of epithelial-mesenchymal transition (EMT) (*Gonzalez et al., 2018*; *Lu et al., 2011*; *Qian and Pollard, 2010*; *Wan et al., 2014*). The specific discrete events in the cancer cell that modulate the tumor immune microenvironment and enable metastasis are not clear.

The *MYC* oncogene is a transcription factor that is one of the most commonly activated oncogenes in the pathogenesis of many types of human cancer including HCC (*Schaub et al., 2018*; *Dang, 2012*; *Gabay et al., 2014*). Previously, we used the Tet System to generate a conditional transgenic mouse model for *MYC*-induced HCC that we and others have used to study mechanisms of oncogene addiction (*Settleman, 2012*) and identify potential therapies (*Dhanasekaran et al., 2018*; *Kapanadze et al., 2013*; *Ma et al., 2016*; *Shachaf et al., 2004*). But murine *MYC*-driven HCC do not metastasize. *Twist1* is a transcription factor that is important during embryogenesis for normal cellular migration (*Lee et al., 1999*; *Thisse et al., 1987*). *Twist1* has been shown to be an important gene product that can enable mouse and human tumor cell lines to acquire the ability to metastasize associated with EMT (*Thiery et al., 2009*; *Xu et al., 2017*). Here we used the Tet System to conditionally express *Twist1* in combination with *MYC* to show that their co-expression leads to widely metastatic and invasive HCC.

We use this powerful in vivo model to uncover a surprising mechanism by which *MYC* and *Twist1* drive metastasis. Cancer cell-intrinsic properties like proliferation, apoptosis or invasiveness were not

different between the non-metastatic *MYC*-HCC and the metastatic *MYC*/*Twist1*-HCC. Instead, metastatic progression was dependent on the ability of *MYC* and *Twist1* to dramatically reprogram the tumor innate immune microenvironment. Together, *MYC* and *Twist1* induce the cancer cell to secrete cytokines like Ccl2 and Il13 that lead to recruitment and polarization of macrophages respectively, thus causing metastasis. Systemically, administering Ccl2 and Il13 is sufficient to cause metastasis of *MYC*-HCC and, conversely blocking these specific cytokines profoundly inhibits metastasis in *MYC*/*Twist1* HCC. Our results are broadly generalizable to 33 different human cancers and predict invasive cancer in a pilot clinical study.

## Results

### Twist1 induces spontaneous metastatic progression of MYC-driven HCC in vivo

We first generated a transgenic mouse using the Tet system that conditionally expresses *Twist1* in a liver specific manner (LAP-tTA/TRE-*Twist1*/Luc). We crossed TRE-*Twist1*/Luc mice which harbored the *Twist1* and firefly luciferase (*luc*) genes under the control of a bidirectional tetracycline responsive element (TRE), with the LAP-tTA mice which contain the tetracycline-controlled transactivator protein (tTA) driven by the liver-enriched activator protein (LAP) promoter (*Tran et al., 2012*). *Twist1* transgenic mice (LAP-tTA/TRE-*Twist1*/Luc) exhibited no disease nor gross or microscopic pathology for as long as 18 months of observation thus demonstrating that *Twist1* did not play a role in autochthonous tumorigenesis when overexpressed in the liver (*Figure 1—figure supplement 1a*).

To examine the influence of *Twist1* on tumor progression, LAP-tTA or LAP-tTA/TRE-*Twist1*/Luc mice were crossed with TRE-*MYC* (*Shachaf et al., 2004*) (*Figure 1a*) to generate transgenic mice that inducibly expressed *MYC* alone (*MYC* mice) or co-expressed *MYC*, *Twist1* and luciferase (Luc) in a liver-specific manner (*MYC*/*Twist1* mice) (*Figure 1b*). We induced transgene expression in adult mice at 6 weeks of age (*Figure 1b*). In vivo, Twist1 transgene expression was confirmed to be confined to the liver by measuring the luciferase reporter by bioluminescence imaging (BLI) (*Figure 1c*). We followed in vivo tumor progression with serial cross-sectional imaging. Both *MYC* and *MYC*/*Twist1* mice were observed to develop multifocal liver cancer, while only *MYC*/*Twist1* mice developed lung metastases (*Figure 1d*). *MYC*/*Twist1* mice were moribund with HCC sooner and had a median survival of 25 months compared to 32 months in *MYC* mice (p<0.001, *Figure 1e*). *MYC* mice rarely exhibited metastasis even after extended observation (2%, n = 50, *Figure 1f*); whereas, *MYC*/*Twist1* mice regularly exhibited rapid onset of metastasis with high penetrance (90%) -metastases to the lungs (70%), peritoneum (60%) and lymph nodes (20%) (n = 50, *Figure 1f*). Thus, *Twist1* combined with *MYC* expression in liver cells elicits HCC metastasis.

A simple explanation for our results is that *Twist1* was inducing more rapid onset and thereby progression of tumorigenesis. Against this possibility, the tumor burden in the liver was not statistically different between *MYC* and *MYC*/*Twist1* mice (*Figure 1g*). Also, there was no difference in the gross or microscopic appearance of *MYC*- and *MYC*/*Twist1*-HCC (*Figure 1h–i*). The *MYC*- and *MYC*/*Twist1*-HCC tumors were confirmed to be HCC by a pathologist and by expression of hepatocyte marker glutamine synthetase (*Figure 1—figure supplement 1b*). We considered that *Twist1* could be influencing *MYC* expression levels, but *MYC* levels were similar between the two tumor models, while *Twist1* was only overexpressed in the *MYC*/*Twist1*-HCC (*Figure 1—figure supplement 1c-d*). Tumor cell proliferative index (phospho histone three expression) and apoptosis (cleaved caspase three) between *MYC*- and *MYC*/*Twist1*-HCC were not different (*Figure 1j–1k*). Primary tumor-derived cell lines from *MYC*- and *MYC*/*Twist1*-HCC did not show any difference in migratory capacity (*Figure 1—figure supplement 1e*). Lastly, *Twist1* is a regulator of epithelial-mesenchymal transition (EMT) [26,37], but we did not observe significant differences in the expression of multiple epithelial and mesenchymal markers between *MYC* and *MYC*/*Twist1* tumors (*Figure 1—figure supplement 1f*). Therefore, *Twist1* drives metastasis of *MYC*-induced HCC without affecting primary tumor burden, *MYC* expression, tumor cell proliferation, apoptosis, invasiveness or EMT markers.

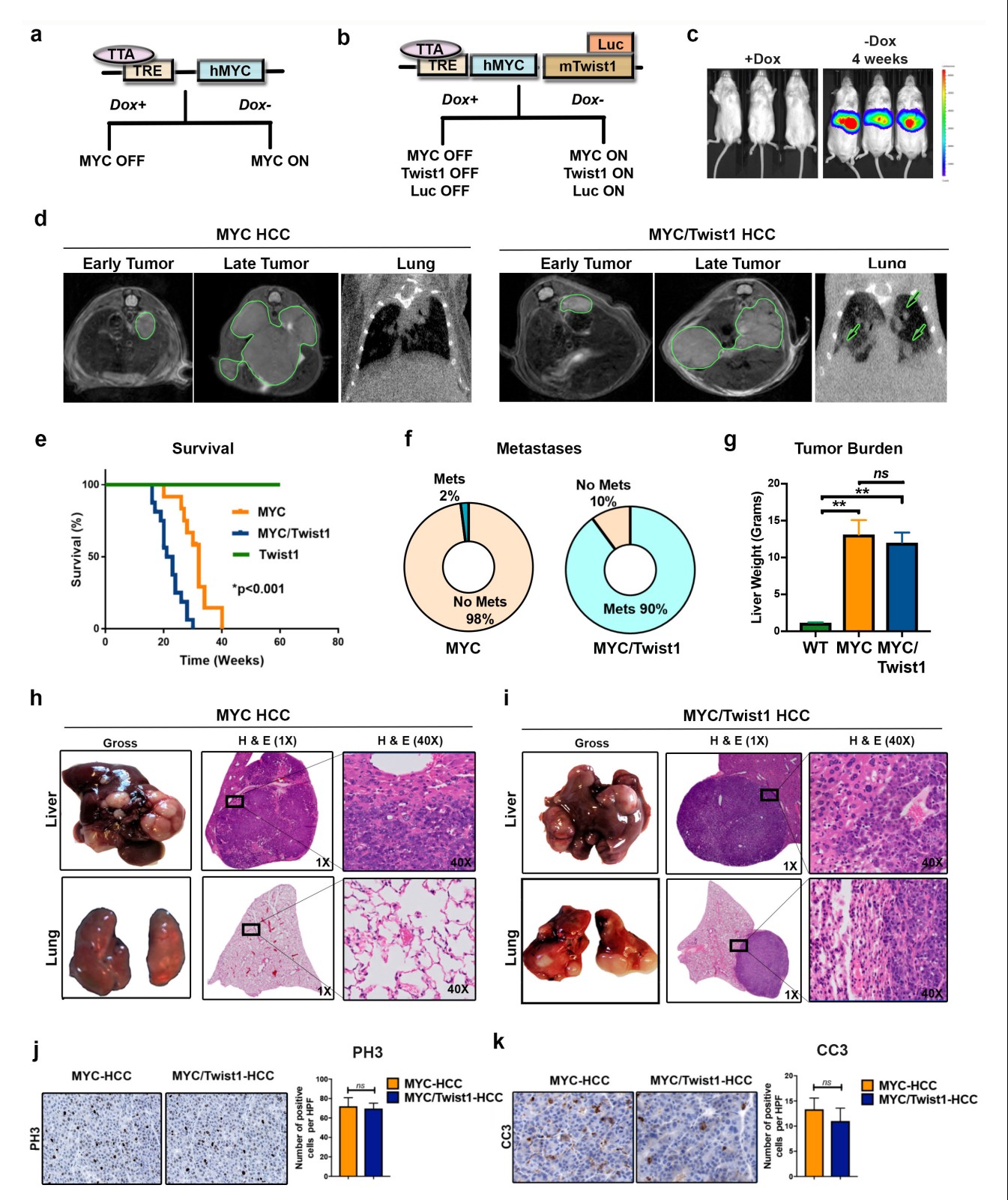

**Figure 1.** Twist1 induces spontaneous metastatic progression of MYC driven HCC in vivo. (**a**) Mouse model of *MYC* induced HCC where *MYC* is under the control of a tetracycline responsive element (TRE) which contain the tetracycline-controlled transactivator protein (tTA) driven by the liver-enriched activator protein (LAP). Doxycycline (Dox) can be used to inactivate oncogene expression in adult mice. (**b**) Mouse model of *MYC/Twist1*-induced HCC which inducibly co-expressed *MYC*, *Twist1* and firefly luciferase in a hepatocyte specific manner. (**c**) Bioluminescent imaging (BLI) confirms in vivo rapid

*Figure 1 continued on next page*

*Figure 1 continued*

induction of oncogenes by demonstrating liver specific luciferase expression upon withdrawal of Dox. (d) Serial cross sectional imaging of *MYC*- (n = 10) and *MYC/Twist1*-HCC (n = 10) using MRI scan for the abdomen and CT scan for the lungs demonstrate step-wise tumor progression. Both *MYC* and *MYC/Twist1* mice develop multifocal liver tumors but only the latter develops lung metastases. (e) Kaplan Meier survival curves show that *MYC/Twist1* mice (n = 16) had significantly shorter survival than *MYC* mice (n = 12) (**p<0.01) while *Twist1* transgenic mice (n = 10) remained healthy. (f) Pie charts show incidence of metastasis in *MYC*- (n = 50) and *MYC*/Twist1 transgenic mice (n = 50). (g) Comparison of liver weights between *MYC* (n = 3) and *MYC/Twist1* tumor bearing mice (n = 4) and control mice (n = 3) which were kept on Dox throughout (**p<0.01). (h) Gross and histopathologic appearance of tumors in *MYC* transgenic model confirming HCC. Lungs do not show any metastases. (i) Representative images showing *MYC/Twist1*-HCC have histologic appearance of HCC and lung histology shows metastatic disease. (j) Representative images from Immunohistochemistry to show phospho histone three (PH3) expression in *MYC*- and *MYC/Twist1*-HCC with quantification of IHC staining. (k) Representative images from Immunohistochemistry to show Cleaved Caspase three (CC3) expression in *MYC*- and *MYC/Twist1*-HCC with quantification of IHC staining.

The online version of this article includes the following figure supplement(s) for figure 1:

**Figure supplement 1.** *Twist1* drives metastasis of *MYC*-induced HCC by non-cell-autonomous mechanisms.

## MYC and Twist1 cooperate to remodel the tumor immune microenvironment

The influence of *Twist1* on global gene expression was measured in *MYC*- and *MYC/Twist1*-HCC (n = 5) using next generation sequencing (NGS) based RNA sequencing. Through unsupervised hierarchical clustering using principal component analysis (PCA), *MYC*- and *MYC/Twist1*-HCC were found to have overall distinct, non-overlapping expression profiles that clustered separately (*Figure 2a*). A comparative analysis identified 514 genes (220 up and 294 down) that were differentially expressed between *MYC*-HCC and *MYC/Twist1*-HCC (p<0.001, q < 0.05, fold change ≥2) (*Figure 2b*, *Supplementary file 1*). Functional pathway analysis revealed the top biological processes upregulated in *MYC/Twist1*-HCC involved inflammatory responses including leukocyte infiltration, myeloid cell and granulocyte recruitment (*Figure 2b*, *Figure 2—figure supplement 1a*, *Supplementary file 2*). CIBERSORT (*Newman et al., 2015*) identified M2 macrophages to be significantly enriched in *MYC/Twist1* tumors, of the 22 immune subsets analyzed (*Figure 2c*). *MYC/Twist1*-HCC exhibited a 15-fold shift in the ratio of M2 to M1 macrophages when compared to *MYC* tumors (*Figure 2c*). No significant differences in other major immune compartments were seen including- B cells, T cells, NK cells, dendritic cells, neutrophils, or mast cells (*Figure 2d*). Increased macrophage infiltration in *MYC/Twist1* primary and metastatic tumors (*Figure 2e*) with no change in neutrophils or CD4 T cells infiltration was confirmed by IHC (*Figure 2—figure supplement 1b*). TAMs isolated from primary *MYC/Twist1*-HCC had increased macrophages of the M2 phenotype (Cd206$^{High}$/Arg1$^{High}$), more specifically a M2a phenotype (*Figure 2f*). Induction of *MYC* and *Twist1* expression is associated with tumor initiation and rapid onset of macrophage infiltration in early tumors which increases during tumor progression. Conversely, the inactivation of *MYC* and *Twist1* in tumors shows rapid and complete tumor regression in all observed mice (n = 20) within 2 weeks (*Figure 2—figure supplement 1c*) and also prompt exodus of macrophages (*Figure 2g–2h*).

TAMs have been shown to increase the migratory capacity of tumor cells (*Lin et al., 2001*). Conditioned media from TAMs isolated from *MYC/Twist1*-HCC but not *MYC*-HCC increased the invasiveness of both *MYC*- and *MYC/Twist1*-HCC tumor cells in vitro (*Figure 3a–3c*).

To discriminate the independent roles of *MYC* and *Twist1* in metastasis we developed cell lines where we could modulate the expression of *MYC* and *Twist1* separately. Primary cell lines from *MYC/Twist1* HCC were retrovirally transduced with *MYC* and/or *Twist1*, such that upon inactivation of transgene expression with Doxycycline, they now constitutively expressed *MYC* and/or *Twist1* (*Figure 3d*). We confirmed that treatment of these cell lines with doxycycline resulted in the continued expression of constitutive *MYC* and/or *Twist1* by qPCR (*Figure 3—figure supplement 1a-b*). We observed that the inactivation of either *MYC* or *Twist1* abrogated the ability of the cells to develop lung metastasis when injected intravenously in NOD scid gamma (NSG) mice, while cells expressing both *MYC* and *Twist1* led to development of extensive lung metastasis with prominent macrophage infiltration (*Figure 3e–3f*, *Figure 3—figure supplement 1c*). Thus, *MYC* and *Twist1* cooperate, and are both required to induce metastasis of HCC by a macrophage dependent mechanism.

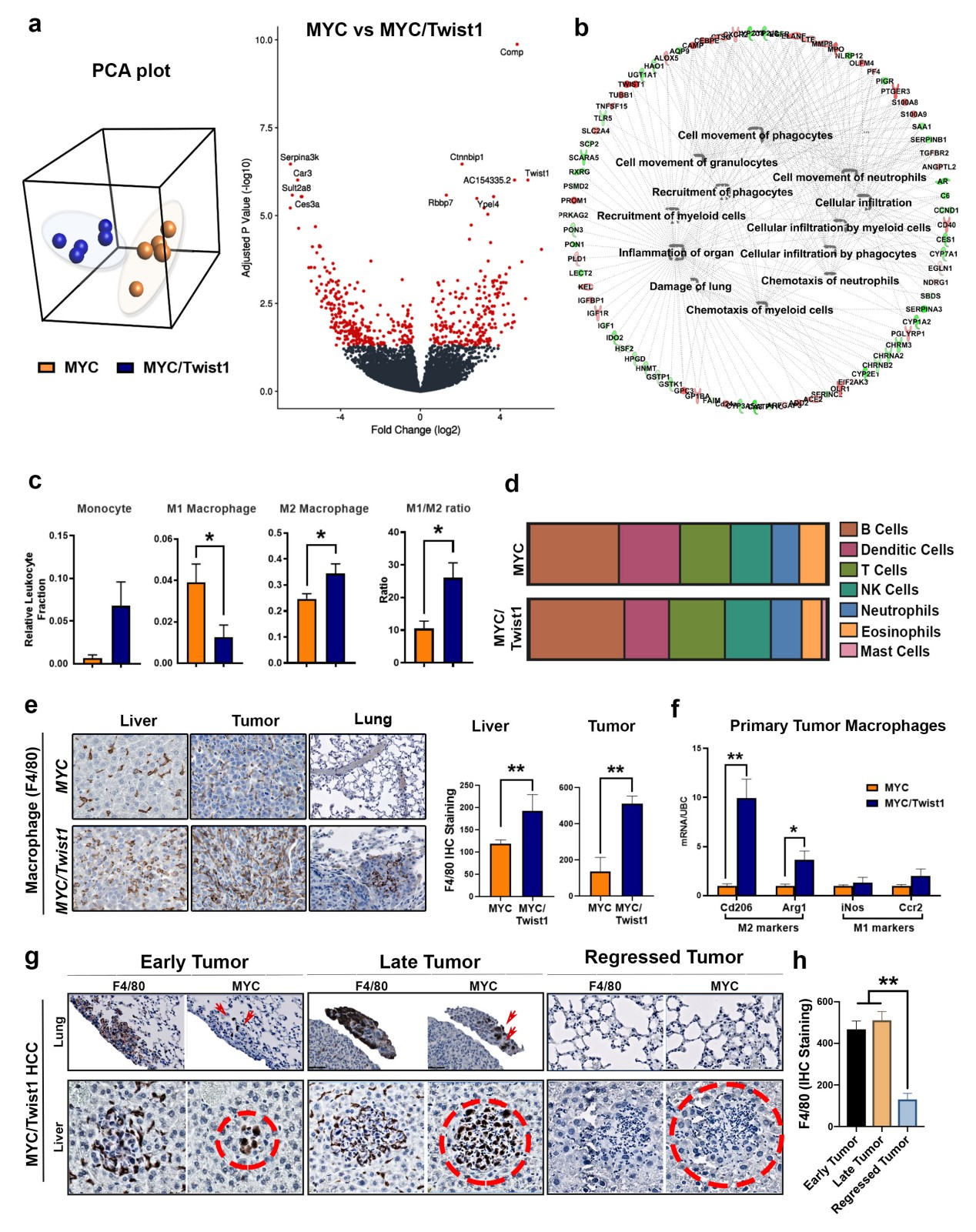

**Figure 2.** MYC and Twist1 cooperate to remodel the tumor immune microenvironment. (**a**) Principal component analysis (PCA) showed that *MYC*-(n = 5) and *MYC/Twist1*-HCC (n = 5) overall had distinct, non-overlapping expression profiles. Volcano plot shows comparative analysis of differentially expressed genes between the *MYC*-HCC and *MYC/Twist1*-HCC. (**b**) Ingenuity pathway analysis of differentially expressed genes between *MYC/Twist1*-(n = 5) and *MYC*-HCC (n = 5) used to identify top biological processes upregulated in *MYC/Twist1*-HCC. (**c**) Comparison of relative percentage of

*Figure 2 continued on next page*

*Figure 2 continued*

monocyte and macrophage subpopulations, derived using CIBERSORT analysis, between *MYC*- (n = 5) and *MYC/Twist1*-HCC (n = 5) (*p<0.05). (**d**) Comparison of relative abundance of major immune subsets between *MYC*- (n = 5) and *MYC/Twist1*-HCC (n = 5) (p=ns). (**e**) Representative images from Immunohistochemistry staining for F4/80 in *MYC* and *MYC/Twist1* normal liver (n = 4), primary tumor (n = 4) and lung (n = 4) with quantification in bar graph (*p<0.05, **p<0.01). (**f**) Macrophages were isolated from primary tumors and expression level of M2 markers (Cd206, Arg1) and M1 markers (iNos, Ccr2) was compared between *MYC*- and *MYC/Twist1*-HCC (*p<0.05). (**g**) Representative images from IHC staining for *MYC* and F4/80 in *MYC/Twist1* HCC in temporal sequence from early tumor (n = 5) to later tumor (n = 5) and tumor regression (n = 5) upon oncogene inactivation. (**h**) Quantification of F4/80 staining in early tumor vs late tumor vs regressed *MYC/Twist1*-HCC (**p<0.01).

The online version of this article includes the following figure supplement(s) for figure 2:

**Figure supplement 1.** Tumor Microenvironment changes in MYC/Twist1 HCC.

## Tumor associated macrophages are required for Twist1 to induce metastasis

We determined if TAMs are required for *Twist1* to drive metastasis on *MYC*-HCC in vivo. Primary tumor-derived cell lines which conditionally express *MYC* or *MYC/Twist1* (*Figure 4—figure supplement 1a*) were re-introduced in vivo either by orthotopic transplantation into the liver or intravenous injection. Orthotopic implantation (*Figure 4a*) of *MYC/Twist1*- but not *MYC*-HCC tumor cells in NSG mice led to pulmonary and intrahepatic metastases with extensive macrophage infiltration (*Figure 4b–c*, *Figure 4—figure supplement 1b*). Macrophage depletion with clodronate liposomes (*Moreno, 2018*) but not control liposomes, in mice orthotopically transplanted with *MYC/Twist1*-HCC had reduced intrahepatic (p=0.0006, FC 4.4) and lung metastases (p<0.0001, FC 8.8) (*Figure 4e–f*). Quantification of BLI signal at the end of treatment did not show statistical difference in densitometry between control treated and clodronate treated mice (*Figure 4e*). Note, clodronate was confirmed to remove macrophages but not affect tumor cells (*Figure 4—figure supplement 1c-d*). A reduction in the number of macrophages was confirmed by IHC for F4/80 in normal liver, tumor and lungs (p<0.001) (*Figure 4—figure supplement 1e-f*).

To evaluate if macrophages are required for the colonization step of metastasis, we used the lung trap assay. Intravenous injection of *MYC/Twist1*-HCC but not *MYC*-HCC cells resulted in pulmonary metastases associated with macrophage infiltration (*Figure 4g–i*, *Figure 4—figure supplement 1g*). Clodronate depletion of macrophages, almost completely abrogated pulmonary metastasis (p=0.0003, FC 4.3) (*Figure 4j–4l*). Therefore, *Twist1* elicits metastasis of *MYC*-HCC and promotes the invasiveness and colonization of metastases by a macrophage-dependent mechanism.

## MYC and Twist1 reprogram the crosstalk between cancer cells and macrophages

To evaluate if cytokines secreted by cancer cells mediate *MYC* and *Twist1* driven macrophage recruitment and polarization, we evaluated the impact of tumor cell derived conditioned media on non-polarized macrophage cell lines. Conditioned media derived from *MYC/Twist1*- but not *MYC*-HCC cells (*Figure 5a*) was sufficient to promote the migration of macrophages towards cancer cells (*Figure 5b*). Conditioned media from *MYC/Twist1*- but not *MYC*-HCC cells was able to elicit changes in the morphology of macrophages to resemble M2 phenotype (*McWhorter et al., 2013*) (*Figure 5c*) and increased expression of M2 markers: *Cd206, Arg1* and *Cc3cr1* (*Figure 5c*), but not M1 markers: iNos, Ccr2, Ifnar2 (*Figure 5—figure supplement 1a*). A multiplex ELISA for 38 cytokines was performed (*Figure 5d*) identifying that *MYC/Twist1*-HCC cells had increased secretion of cytokines: Il13, Ccl2, Ccl5, Ccl7 and Cxcl1 (p<0.05; Fold change ≥2, mean ≥20 ng/ml) (*Figure 5e*, *Figure 5—figure supplement 1b*). These five cytokines were confirmed to be transcriptionally upregulated in *MYC/Twist1*- vs. to *MYC* cells by qPCR (*Figure 5e*).

The role of individual cytokines in macrophage recruitment was assessed. Antibodies that neutralize Ccl2, Ccl5, Ccl7 or Cxcl1 inhibited the ability of conditioned media from *MYC/Twist1*-HCC to promote migration of macrophages (*Figure 5f–h*). Neutralization of Ccl2 decreased migration 7-fold (p<0.0001), Ccl5 1.2-fold (p=0.001), Ccl7 1.8-fold (p=0.001) and Cxcl1 2.1-fold (p=0.0002). Neutralizing Il13 (p=*ns*, FC 1.2) did not affect macrophage migration. Also, we found that the neutralization of Ccl2 inhibited the recruitment of macrophages into *MYC/Twist1*-spheroids by 3D culture (*Figure 5—figure supplement 1c*).

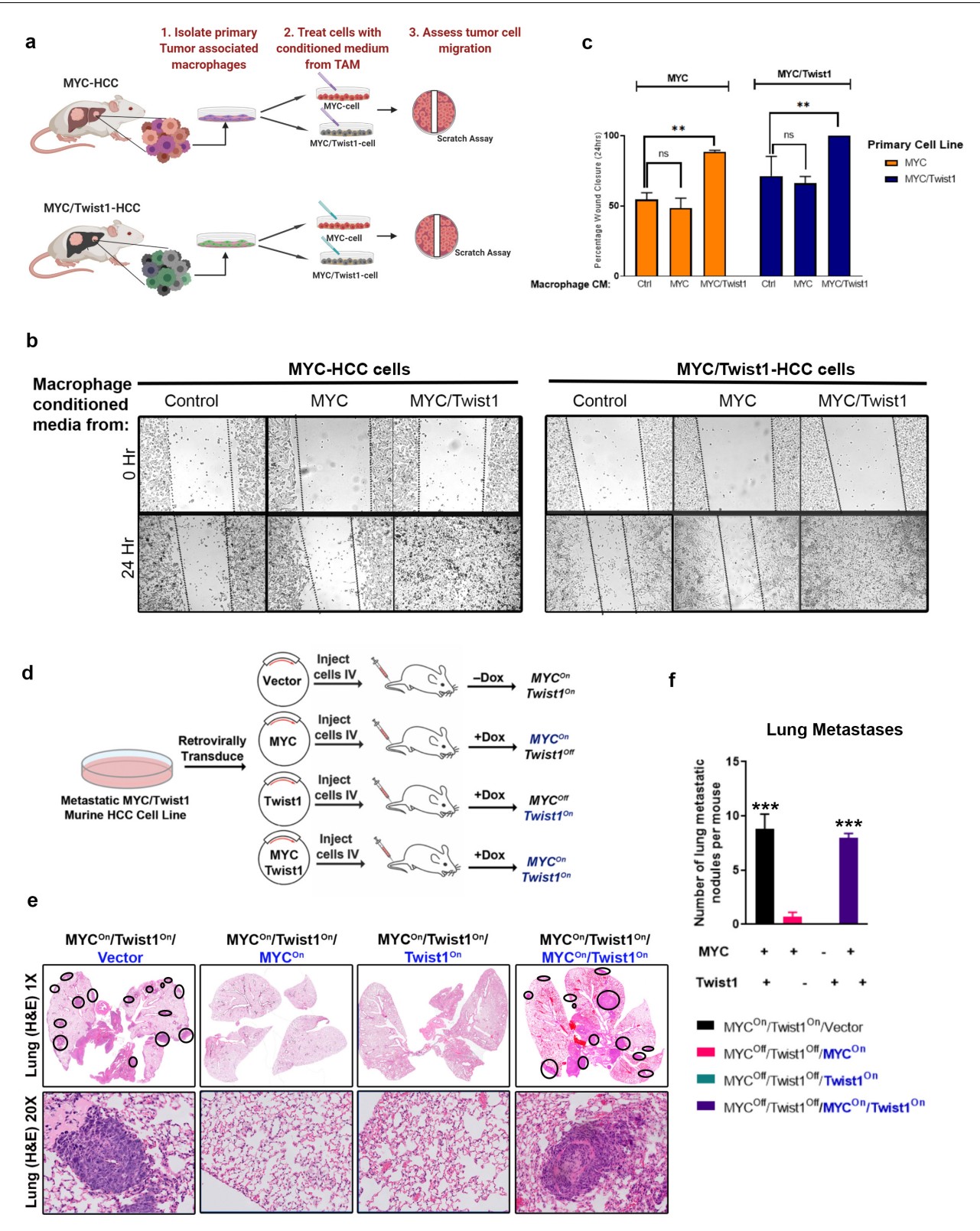

**Figure 3.** Coordinate expression of both MYC and Twist1 are necessary for inducing metastasis. (**a**) Schematic of the experiment to extract tumor associated macrophages from the primary tumors of *MYC*- (n = 5) and *MYC/Twist1*-HCC (n = 5). Conditioned media from macrophages extracted from *MYC*-HCC (n = 3 biological replicates with three technical replicates each) or *MYC/Twist1*-HCC tumors (n = 3 biological replicates with three technical replicates each) was used to treat either *MYC*- or *MYC/Twist1*-HCC cells. (**b**) Wound healing assay was performed in *MYC*- or *MYC/Twist1*-HCC cells

*Figure 3 continued on next page*

*Figure 3 continued*

which were treated with conditioned media derived from TAMs isolated from primary *MYC*- or *MYC/Twist1*-HCC. (c) Bar graphs show quantification of percentage wound closure at 24 hr (**p<0.01). (d) Schematic showing generation of cell lines which constitutively express *MYC* and/or *Twist1* (blue font) upon transgene inactivation with Dox treatment. (e) H and E (1X and 20X) of lungs of mice injected with the four different cell lines as shown in their individual titles. (f) Quantification of lung metastatic burden upon intravenous (IV) injection of cell lines expressing either *MYC* or *Twist1* alone or both (***p<0.001).

The online version of this article includes the following figure supplement(s) for figure 3:

**Figure supplement 1.** MYC and Twist1 cooperate to induce recruitment of tumor associated macrophages.

Next, the role of cytokines on macrophage polarization was determined (*Figure 5h*). We found that neutralization of Il13 blocked M2 polarization by 50-fold reduction in Cd206 expression (p<0.0001) and 7-fold decrease in Arg1 (p<0.0001) without any change in M1 markers. Neutralization of Ccl5 led to a 4-fold reduction in Cd206 (p<0.001) and 1.1-fold decrease in Arg1 (p<0.05) and Cxcl1 led to 2.4-fold decrease in Cd206 (p<0.001) without significant change in Arg1 (p=ns) (*Figure 5i*). Conversely, adding the cytokines Il13, but not Ccl2, to co-cultured *MYC*-HCC cells increased M2 markers expression (*Figure 5—figure supplement 1d*).

*MYC* and *TWIST1* are both transcription factors, so we evaluated if they epigenetically regulated *CCL2* and *IL13* expression. We identified *MYC* and *TWIST1* promoter binding upstream of human *CCL2* and *IL13* protein-coding genes in Gene Transcription Regulation Database (GTRD), a meta-analysis of Chip-seq experiments (Dreos et al. 2017; *Yevshin et al., 2019*) (*Figure 5—figure supplement 1e*). Both *MYC* and *Twist1* demonstrated binding at multiple sites in the promoter regions of *CCL2* and *IL13* in the ChIP-seq data from several different cancer cell lines (*Figure 5—figure supplement 1e*). We also looked for *MYC* and *Twist1* promoter binding sites in mouse *Ccl2* and *Il13* promoters using motif finding analysis of the public data from JASPAR (*Bryne et al., 2008*) and Eukaryotic promoter database (EPD) (Dreos et al. 2017). Again, we found multiple potential *MYC* and *Twist1* transcription factor binding sites for both *Ccl2* and *Il13* (*Figure 5—figure supplement 1f*). These data suggest that *MYC* and *Twist1* cooperate to transcriptionally regulate expression of *Ccl2* and *Il13* in the cancer cells.

## Ccl2 and Il13 are sufficient and required for metastasis of HCC

We examined if Ccl2 and/or Il13 are sufficient to elicit metastasis in vivo. Orthotopic transplants of non-metastatic *MYC*-HCC in NSG mice were treated with either PBS (control), or with recombinant Ccl2 alone or Il13 alone or their combination for 4 weeks (*Figure 6a*). Control mice did not develop metastatic nodules even though scattered, single cells were found in the lungs (*Figure 6b*). No mice treated with Ccl2 alone (p=0.390, FC = 3) or Il13 alone (p=0.99, FC = 1) exhibited intrahepatic or pulmonary metastases (*Figure 6c, e and g*). All orthotopic *MYC*-HCC mice treated with the combination of Ccl2 and Il13 developed intrahepatic metastases (p=0.008, FC 2.5) and multifocal pulmonary metastases (p=0.02, FC = 104.4) (*Figure 6c, e and g*). Hence, both Ccl2 and Il13 are sufficient to elicit *MYC*-HCC to metastasize, even if *Twist1* is not expressed in the tumor cells.

We evaluated if treatment with cytokines altered tumor macrophage infiltration. The combined treatment with Ccl2 and Il13 increased macrophage infiltration at both the orthotopic primary tumor in the liver and pulmonary metastases (p<0.0001) (*Figure 6e and h*), with enrichment of M2-like Cd206+ TAMs (*Figure 6—figure supplement 1a-b*). Treatment with Ccl2 alone increased macrophage recruitment 1.8-fold (p=0.001) but did not induce metastasis (*Figure 6c, d, f and g*). On the other hand, treatment with Il13 alone did not increase either macrophage infiltration or induce metastasis (*Figure 6e and h*). Also, in vivo treatment of *MYC*-HCC with combination of Ccl2 and Il13 stimulated angiogenesis. The most significant increase in angiogenesis, as assessed by Cd31 IHC staining, was noted in *MYC*-HCC tumors in mice treated with Ccl2 and Il13 (*Figure 6—figure supplement 1c*). As a negative control, treatment with recombinant Il4, a cytokine which was not increased in *MYC/Twist1*-HCC was performed. Il4 did not elicit either macrophage recruitment or increase metastasis (*Figure 6—figure supplement 1d-e*). Thus, combination of Ccl2 and Il13 induces metastasis of *MYC*-HCC associated with macrophage recruitment and polarization.

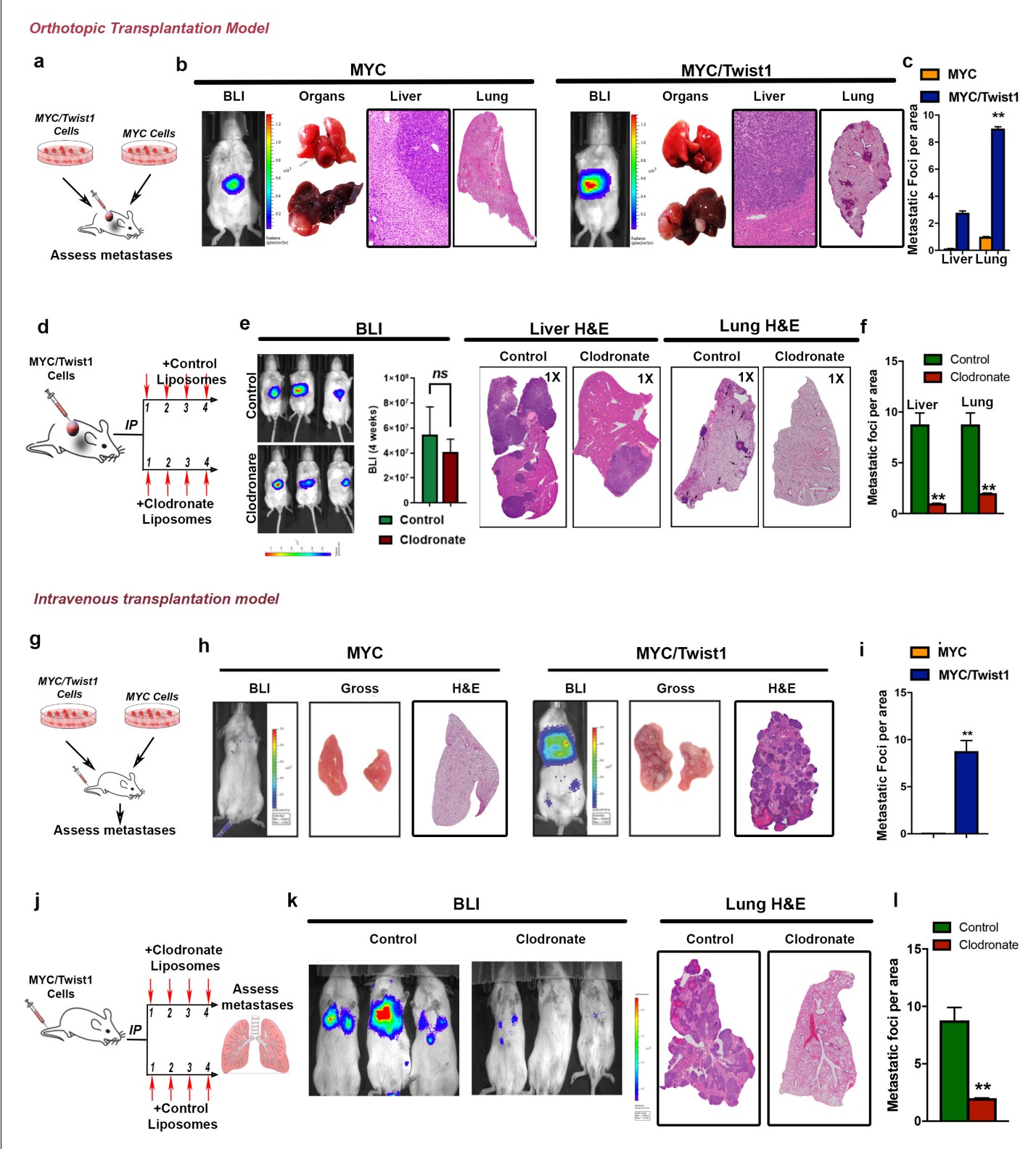

**Figure 4.** Tumor associated macrophages are required for Twist1 to induce metastasis of MYC-HCC. (a) Experimental scheme- *MYC* and *MYC/Twist1* cells were implanted orthotopically and metastatic burden in liver and lung assessed after 4 weeks. (b) Representative BLI imaging, gross organ appearance, histopathology of liver (10X) and lungs (1X) from mice orthotopically implanted with *MYC* (n = 5) and *MYC/Twist1* cells (n = 5). (c) Comparative quantification of liver and lung metastatic burden between *MYC* (n = 5) and *MYC/Twist1* orthotopic HCC (n = 5). (**p<0.01). (d) Experimental model of orthotopic *MYC/Twist1*-HCC treatment either with control liposomes or clodronate liposomes for 4 weeks for macrophage
*Figure 4 continued on next page*

*Figure 4 continued*

depletion. (e) Representative BLI imaging, BLI quantification, gross organ appearance, histopathology of liver and lungs from *MYC/Twist1* orthotopic HCC bearing mice treated with either control liposomes (n = 5) or clodronate liposomes (n = 4). (f) Comparative quantification of liver and lung metastatic burden between *MYC/Twist1* orthotopic HCC bearing mice treated with either control liposomes (n = 5) or clodronate liposomes (n = 4) (**p<0.01). (g) Experimental scheme- *MYC* and *MYC/Twist1* cells were injected intravenously and metastatic burden in lung assessed after 4 weeks. (h) Representative BLI imaging, gross organ appearance, histopathology of lungs from mice intravenously injected with *MYC* (n = 5) and *MYC/Twist1* cells (n = 4). (i) Comparative quantification of lung metastatic burden between *MYC* (n = 5) and *MYC/Twist1* intravenously injected HCC (n = 4). (**p<0.01). (j) Experimental model of intravenous *MYC/Twist1*-HCC treatment either with control or clodronate liposomes for 3 weeks for macrophage depletion. (k) Representative BLI imaging, and lung histopathology from *MYC/Twist1* intravenous HCC injected mice treated with either control liposomes (n = 4) or clodronate liposomes (n = 5). (l) Comparative quantification of liver and lung metastatic burden between *MYC/Twist1* intravenously injected HCC bearing mice treated with either control liposomes (n = 4) or clodronate liposomes (n = 5) (**p<0.01).

The online version of this article includes the following figure supplement(s) for figure 4:

**Figure supplement 1.** Effect of Macrophage depletion on MYC/Twist1 HCC metastasis.

## Combined inhibition of Ccl2 and Il13 synergistically inhibits metastasis in vivo

We examined if neutralizing antibodies to Ccl2 and/or Il13 influenced metastasis in vivo. Mice with orthotopic transplants of metastatic *MYC/Twist1*-HCC were treated either with control antibody, or anti-Ccl2 antibody alone or anti-Il13 antibody alone or their combination for 4 weeks (*Figure 7a*). We first confirmed that there was no statistical difference in orthotopic primary liver tumor burden between the four groups by quantifying primary tumor volume. Quantification of BLI imaging showed a trend towards decrease but no statistical difference in liver tumor burden between the four groups (*Figure 7b–c*). Control antibody treated mice, as expected, developed multifocal intra-hepatic and pulmonary metastases with extensive macrophage infiltration (*Figure 7c–e*). Inhibition of Ccl2 alone led to 7-fold decrease (p=0.002) and Il13 alone to a 2.5-fold decrease (p=0.03) in lung metastasis (*Figure 7c and e*). Further, treatment with combination of anti-Ccl2 and anti-Il13 antibodies showed synergism, and led to a 12-fold decrease in liver metastases (p=0.0006, *Figure 7c and d*) and 14-fold decrease in lung metastases (p=0.0009) (*Figure 7c and e*).

The combined inhibition of Ccl2 and Il13 was noted to lead to dramatic reduction in macrophage recruitment (p<0.0001, FC 7.1)(*Figure 7c and f*) and polarization to M2-like phenotype (*Figure 7—figure supplement 1a*). The inhibition of Ccl2 alone (p<0.001, FC 4.0), but not Il13 (p=0.435), decreased macrophage recruitment (*Figure 7c and f*). While inhibition of Il13, either alone or in combination with Ccl2, led to loss of M2-like Cd206+ or Arg1+ macrophages in primary or metastatic sites (*Figure 7—figure supplement 1a–b*). Thus, inhibition of Ccl2 and Il13 decrease macrophage recruitment and polarization respectively. As a control, mice bearing *MYC/Twist1*-HCC were treated with anti-Il4 antibody, which we show had no effect on metastasis or macrophage recruitment (*Figure 7—figure supplement 1 c–e*). Therefore, combined inhibition of Ccl2 and Il13 synergistically reduced macrophage recruitment and polarization thus blocking metastases (*Figure 7g*).

## MYC and TWIST1 predict poor prognosis, TAM infiltration and pro-TAM cytokines in 33 human cancers

We examined if *MYC* and *TWIST1* cooperated in human tumorigenesis by examining 9502 human patients with 33 different cancers from a TCGA study (*Tang et al., 2017*) and 144 HCC patients with metastatic HCC (*Ye et al., 2003*). Increased *MYC* and *TWIST1* expression was associated with significantly worse disease-free survival (DFS) (p=$4.3 \times 10^{-10}$) in the pan-cancer cohort (*Figure 8a*). Combined overexpression of *MYC*, *TWIST1*, *CCL2* and *IL13* predicted a slightly worse disease-free survival than *MYC+TWIST1* overexpression alone (p=$2.9 \times 10^{-12}$) (*Figure 8a*). In another cohort of 144 patients with metastatic HCC, combined overexpression of *MYC* and *TWIST1* was associated with significantly worse prognosis than either *MYC* or *TWIST1* alone (*Figure 8b*).

We determined if *MYC* and *TWIST1* in human tumors cooperated to influence the tumor microenvironment through CIBERSORT analysis of 10,366 tumors from the human pan-cancer TCGA study (*Thorsson et al., 2018*). TAMs were the most common infiltrating immune cells in most types of human cancers (*Figure 8—figure supplement 1a*) including HCC (*Figure 8—figure supplement 1b*). Compared to *MYC/TWIST1*[Low] tumors, *MYC/TWIST1*[High] tumors were infiltrated with significantly higher proportion of monocytes (2.9% vs 4.4%; p<0.001) and M2 macrophages (22% vs 26%,

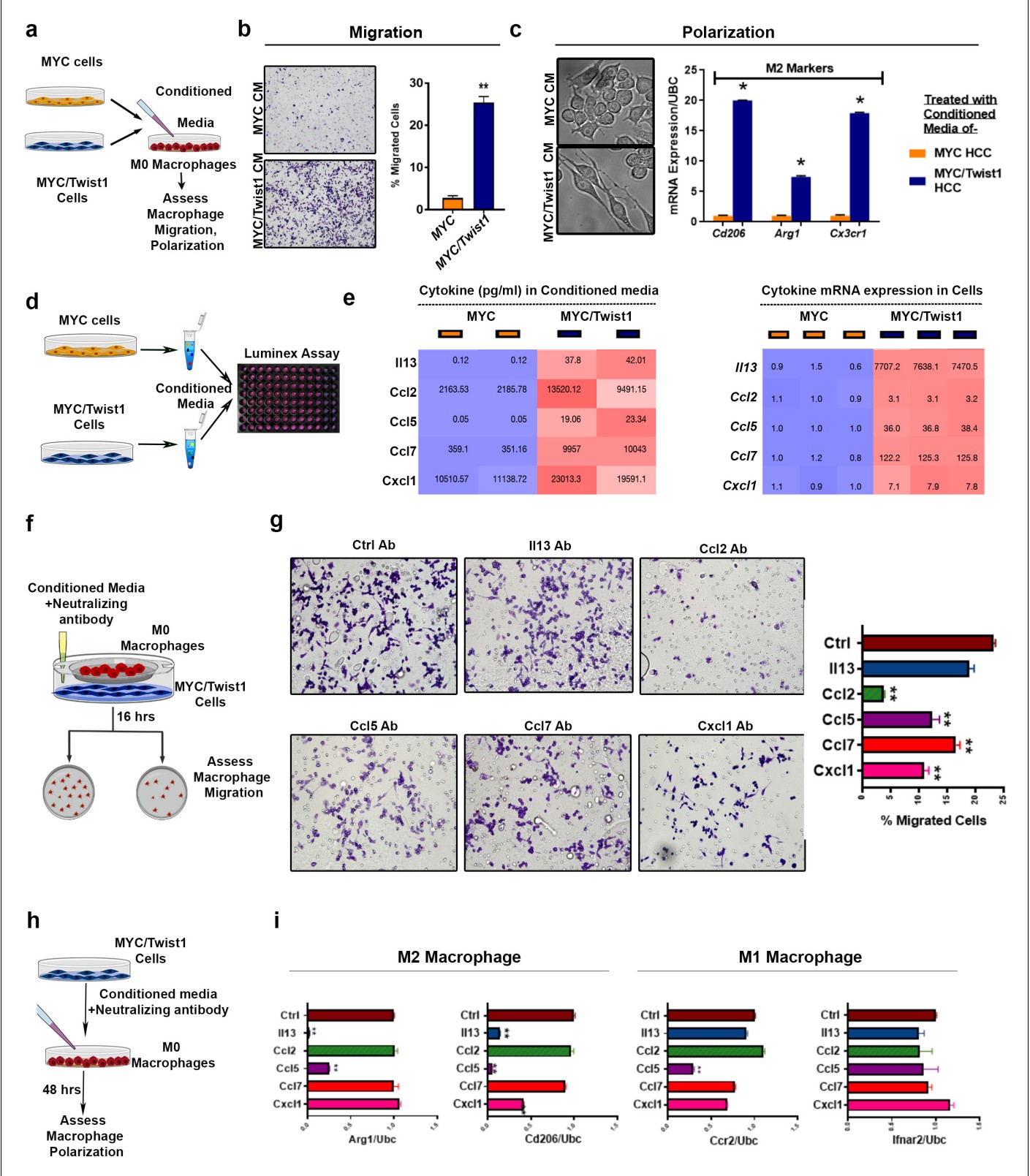

**Figure 5.** MYC and Twist1 reprogram the cytokinome to induce macrophage recruitment and polarization. (a) Experimental scheme- conditioned media (CM) from *MYC* or *MYC/Twist1* cells was used to treat non polarized macrophages for 48 hr. Following that, macrophage migration or polarization was assessed. (b) Transwell macrophage migration across a membrane insert when treated with CM from *MYC*-cells (n = 3 biological replicates, with three technical replicates each) or *MYC/Twist1*-cells (n = 3 biological replicates, with three technical replicates each). Bar graph shows

*Figure 5 continued on next page*

*Figure 5 continued*

quantification of migrated cells. (c) Morphologic appearance of macrophages treated with CM from *MYC* or *MYC/Twist1* cells. Expression of M2 markers (CDd06, Arg1, Cx3cr1) in macrophages treated with CM from *MYC* (n = 3 biological replicates, with three technical replicates each) or *MYC/ Twist1* cells (n = 3 biological replicates, with three technical replicates each). (*p=0.05). (d) Experimental scheme- the CM from *MYC* or *MYC/Twist1*cells were analyzed using Luminex-plate based multiplex ELISA assay. (e) Heatmap showing expression levels of top five differentially secreted cytokines in CM of *MYC* (n = 3 biological replicates, with three technical replicates each) or *MYC/Twist1* cells(n = 3 biological replicates, with three technical replicates each) by ELISA. Second heatmap showing mRNA expression levels of top five cytokines between *MYC* or *MYC/Twist1* cells by qPCR. (f) Experimental scheme- Co-culture of *MYC/Twist1* cells and macrophages separated by a chamber to evaluate chemotaxis of macrophages towards the cancer cells was performed. Neutralizing antibodies to individual cytokines or control antibody were added to the CM of *MYC/Twist1* cells. (g) Transwell chamber migration assay of macrophages in the upper chamber toward the *MYC/Twist1* cells in the lower chamber. *MYC/*Twist cells CM was treated with control antibody or neutralizing antibody to Il13, Ccl2, Ccl5, Ccl7 or Cxcl1 respectively. (n = 3 biological replicates, with three technical replicates each)(**p<0.01). (h) Experimental scheme- CM of *MYC/Twist1* cells treated with control antibody or neutralizing antibody to Il13, Ccl2, Ccl5, Ccl7 or Cxcl1 respectively was added to non-polarized macrophages for 48 hr. (i) Macrophage polarization was assessed by qPCR for M2 markers (ARG1, CD2016) and M1 markers (Ifnar2, Ccr2)(n = 3 biological replicates, with three technical replicates each) (***p<0.001).

The online version of this article includes the following figure supplement(s) for figure 5:

**Figure supplement 1.** *Mechanisms of MYC and Twist1 cooperation.*

p<0.001), while there was a lower proportion of M0 (10% vs 7%, p<0.001) and M1 macrophages (6% vs 4%, p<0.001) (*Figure 8c*). This was also true in the HCC cohort (M2 macrophages 24% vs 28% p=0.001; *Figure 8c*). Moreover, combined *MYC* and *TWIST1* expression strongly correlated with the expression of M2 macrophage related genes in the pan-cancer TCGA data (*Figure 8d*). Increased TAM infiltration in HCC was highly prognostic of overall survival on univariate and multivariate analysis (p=0.01, HR 16.0, *Figure 8—figure supplement 1c-e*). Also, M2-like TAMs were associated with presence of vascular invasion, advanced stage and poor tumor grade (*Figure 8—figure supplement 1f*) in HCC. Further, combined *MYC* and *TWIST1* expression correlated strongly with CCL2 and IL13 in the pan-cancer cohort (p=$1.4 \times 10^{-109}$) (*Figure 8d*). Thus, *MYC* and *TWIST1* predict poor survival, CCL2/IL13 expression and M2-like TAM infiltration in human cancers.

We screened *MYC* and *TWIST1* expression in four human HCC cell lines-Huh7, SNU398, SNU475 and SNU182. We identified Huh7 as a *MYC/TWIST1*$^{Low}$ cell line and SNU398 as *MYC/TWIST1*$^{High}$ cell line (*Figure 8e*). The other two cell lines had intermediate levels of TWIST1. The *MYC/TWIST1*$^{High}$ cells transcriptionally expressed significantly higher levels of CCL2 and IL13 when compared to the *MYC/TWIST1*$^{Low}$ cells (*Figure 8e*). We treated macrophages with conditioned media derived from either *MYC/TWIST1*$^{Low}$ or *MYC/TWIST1*$^{High}$ cells, in vitro, and assessed their effect on macrophage polarization (*Figure 8f*). Conditioned media from *MYC/TWIST1*$^{High}$ cells polarized macrophages to M2-like phenotype while conditioned media from *MYC/TWIST1*$^{Low}$ cells induced a M1-like phenotype (*Figure 8g*). Thus, combined *MYC* and *TWIST1* in human HCC cells lines is associated with CCL2 and IL13 secretion and macrophage M2-like polarization.

Lastly, a prospective clinical study was performed to measure CCL2 and IL13 in human patients with HCC (n = 25) and in patients with cirrhosis of the liver (n = 10). Both CCL2 (p=0.006) and IL13 (p<0.0001), were significantly elevated in the plasma of patients with HCC but not cirrhosis (*Figure 8h*). Increased expression of IL13, but not CCL2, was associated with the presence of multifocal tumors (p=0.04), suggestive of association with aggressive phenotype (*Figure 8i*). Also, IL13 levels were higher in the plasma of patients with HCC with vascular invasion, which is a known indicator of metastatic tumor spread in HCC (p=0.01) (*Figure 8j*). Thus, the plasma levels of the cytokines CCL2 and IL13 are elevated in patients with HCC, and higher IL13 levels predict multifocal and invasive HCC.

## Discussion

We found that *MYC* and TWIST1 drive metastasis by eliciting a transcriptional program in cancer cells that induces cytokines that in turn enable crosstalk between tumor and host, thus eliciting the recruitment and polarization of macrophages (*Figure 9*). To perform our studies, we have generated the first autochthonous transgenic mouse model of HCC metastasis by hepatocyte-specific expression of *MYC* and *Twist1* that will be useful to study tumor progression and identify new therapies. We demonstrate that constitutive expression of both *MYC* and *Twist1* is required for metastasis.

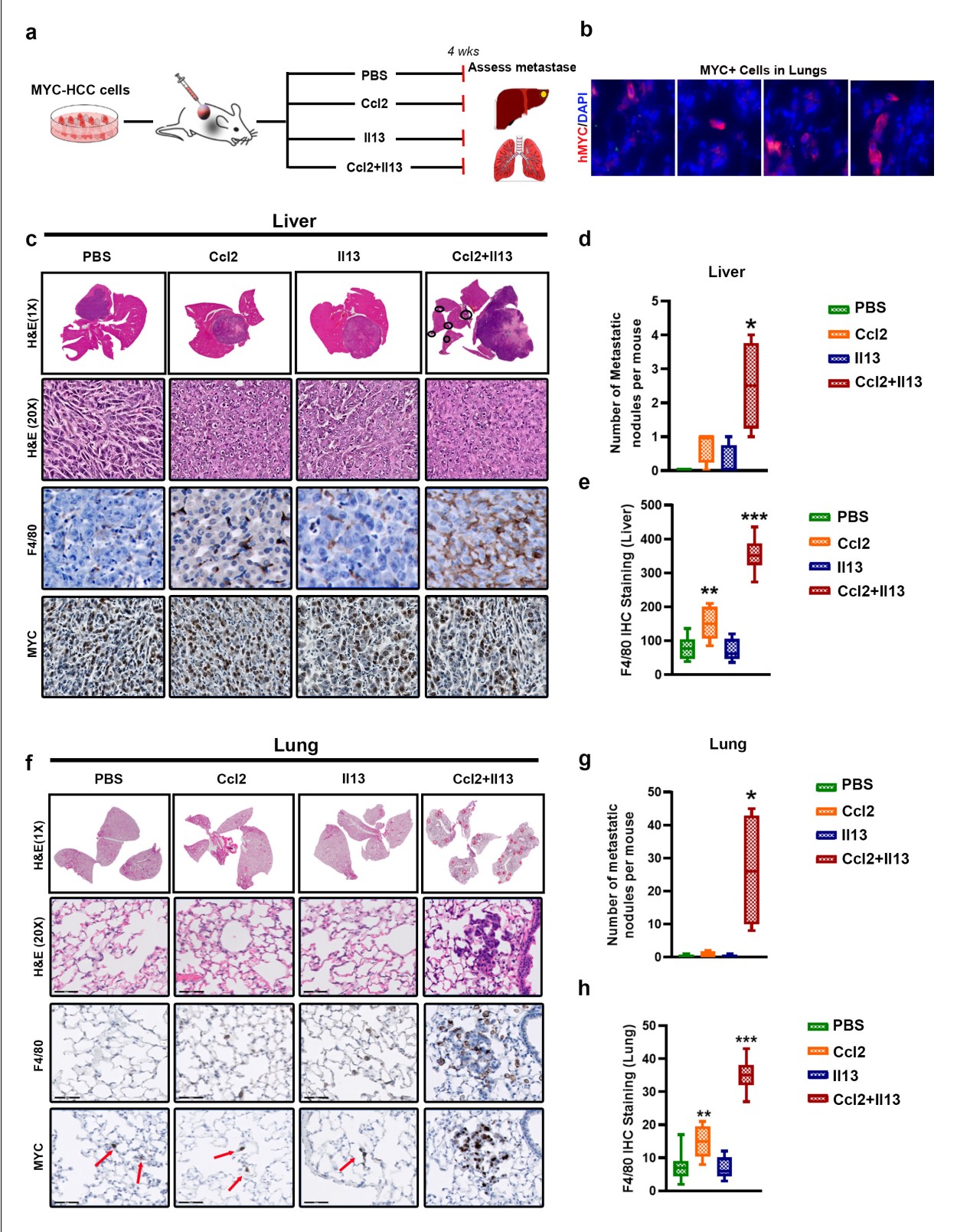

**Figure 6.** MYC and TWIST1 require both Ccl2 and Il13 to promote metastasis. (a) Experimental scheme- Mice orthotopically transplanted with *MYC*-HCC cells were treated either with PBS or Il13 or Ccl2 or Ccl2+Il13 recombinant cytokines for 4 weeks. (b) *MYC* expressing single cells scattered in the lungs of mice orthotopically bearing *MYC*-HCC. (c) Histopathology of liver (1X and 20X) of *MYC*-HCC bearing mice treated with PBS (n = 4) or Ccl2 (n = 4) or Il13 (n = 4) or Ccl2+Il13 (n = 4) and IHC for F4/80 and *MYC* expression in each group. (d) Quantification of number of metastatic nodules in

*Figure 6 continued on next page*

*Figure 6 continued*

the liver control treated or recombinant cytokine treated *MYC*-HCC bearing mice. (*p<0.05). (**e**) Quantification of macrophage infiltration in liver by IHC staining for F4/80 of *MYC*-HCC bearing mice liver treated with PBS (n = 4) or Ccl2 (n = 4) or Il13 (n = 4) or Ccl2+Il13 (n = 4). (**f**) Representative images from histopathology of lung (1X and 20X) of *MYC*-HCC bearing mice treated with PBS or Ccl2 or Il13 or Ccl2+Il13 and IHC for F4/80 and *MYC* expression in each group. (**g**) Quantification of number of metastatic nodules in the lung of control treated or recombinant cytokine treated *MYC*-HCC bearing mice. (**p<0.01). (**h**) Quantification of macrophage infiltration in lung by IHC staining for F4/80 of *MYC*-HCC bearing mice liver treated with PBS (n = 4) or Ccl2 (n = 4) or Il13 (n = 4) or Ccl2+Il13 (n = 4).

The online version of this article includes the following figure supplement(s) for figure 6:

**Figure supplement 1.** MYC and Twist1 induce macrophage polarization and angiogenesis via Il13.

*MYC* and *Twist1* coordinate to regulate a cytokinome, including Ccl2 and Il13, that both are necessary and sufficient to elicit metastasis. Moreover, in 33 different human cancers, expression of both *MYC* and TWIST1 predict survival, expression of CCL2/IL13 and M2-like TAM infiltration. In a prospective clinical analysis, human patients with HCC but not cirrhosis had increased plasma levels of CCL2 and IL13 that was associated with invasive and multifocal disease. We conclude that *MYC* and *TWIST1* are general drivers of metastasis.

We generated a new conditional transgenic mouse model of metastasis that has some features that are complementary to existing model systems (*Gómez-Cuadrado et al., 2017*). Our models enabled us to interrogate tumor and host interactions during malignant progression in an immunocompetent host. Through the Tet system, we were able to elicit inducible combined *MYC* and *Twist1* expression restricted to the hepatocytes. In our model, greater than 90% of the mice predictably developing extrahepatic metastasis. This allowed us to directly compare the non-metastatic *MYC*-HCC and the metastatic *MYC/Twist1*-HCC, to identify specific mechanisms of metastasis. We believe our model will serve as a useful system for developing new therapies that block cancer metastasis.

Our work is the first to suggest that *MYC* and *Twist1* generally cooperate to drive metastasis. By analyzing ChIP-seq data, we found *MYC* and *Twist1* bind to both the promoters of human and mouse *Ccl2* and *Il13* genes. Our results are consistent with a recent report that *Twist1* promotes the invasion of *MYCN* to enhancer sites thus potentiating its pro-proliferative function (*Zeid et al., 2018*). Thus, our results suggest that *MYC* and *Twist1* may contribute to metastasis through a transcriptional mechanism of inducing a cytokinome that activates macrophages. We note that our observations are consistent with a multitude of reports that innate immunity contributes to metastasis (*Kitamura et al., 2015*; *Pollard, 2004*) and *Twist1* contributes to metastasis (*Xu et al., 2017*; *Yang et al., 2004*). However, our work is the first to suggest that *MYC* and *Twist1* together affect transcription in a manner that modulates innate immunity, thereby driving metastasis.

We propose as a possible general explanation for our findings that overexpression of *MYC* and *TWIST1* in a tumor is activating an embryonic program of innate immune cell activation and cellular invasion. *MYC* and *TWIST1* have been reported to cooperate during embryogenesis (*Bellmeyer et al., 2003*; *Rodrigues et al., 2008*). The two transcription factors also have been shown to transcriptionally modulate inflammation during embryogenesis (*Hurlin, 2013*; *Rodrigues et al., 2008*; *Spicer et al., 1996*). These microenvironment changes are required to enable mesodermal cells to migrate to their destination (*Šošić et al., 2003*). Both *MYC* and *TWIST1* are overexpressed in multiple human cancers, suggesting there is a common embryonic transcriptional program they regulate in the embryonic microenvironment which is hijacked by cancer cells (*Hendrix et al., 2007*). Hence, *MYC* and TWIST1 overexpression in cancer may be eliciting tumor invasion by activating embryonic programs that otherwise physiologically enable mesodermal migration.

We identified that *MYC* and *Twist1* generally elicit a cytokinome that includes Ccl2 and Il13 that we show are necessary and sufficient to drive metastasis. Our work is consistent with a prior report that *Twist1* transcriptionally induces Ccl2 in breast cancer cell lines that leads to macrophage recruitment (*Low-Marchelli et al., 2013*). However, our study further demonstrates Ccl2 induced macrophage recruitment alone was not sufficient to cause metastasis in vivo and Il13 induced macrophage polarization plays an essential and complementary role in promoting angiogenesis and metastasis.

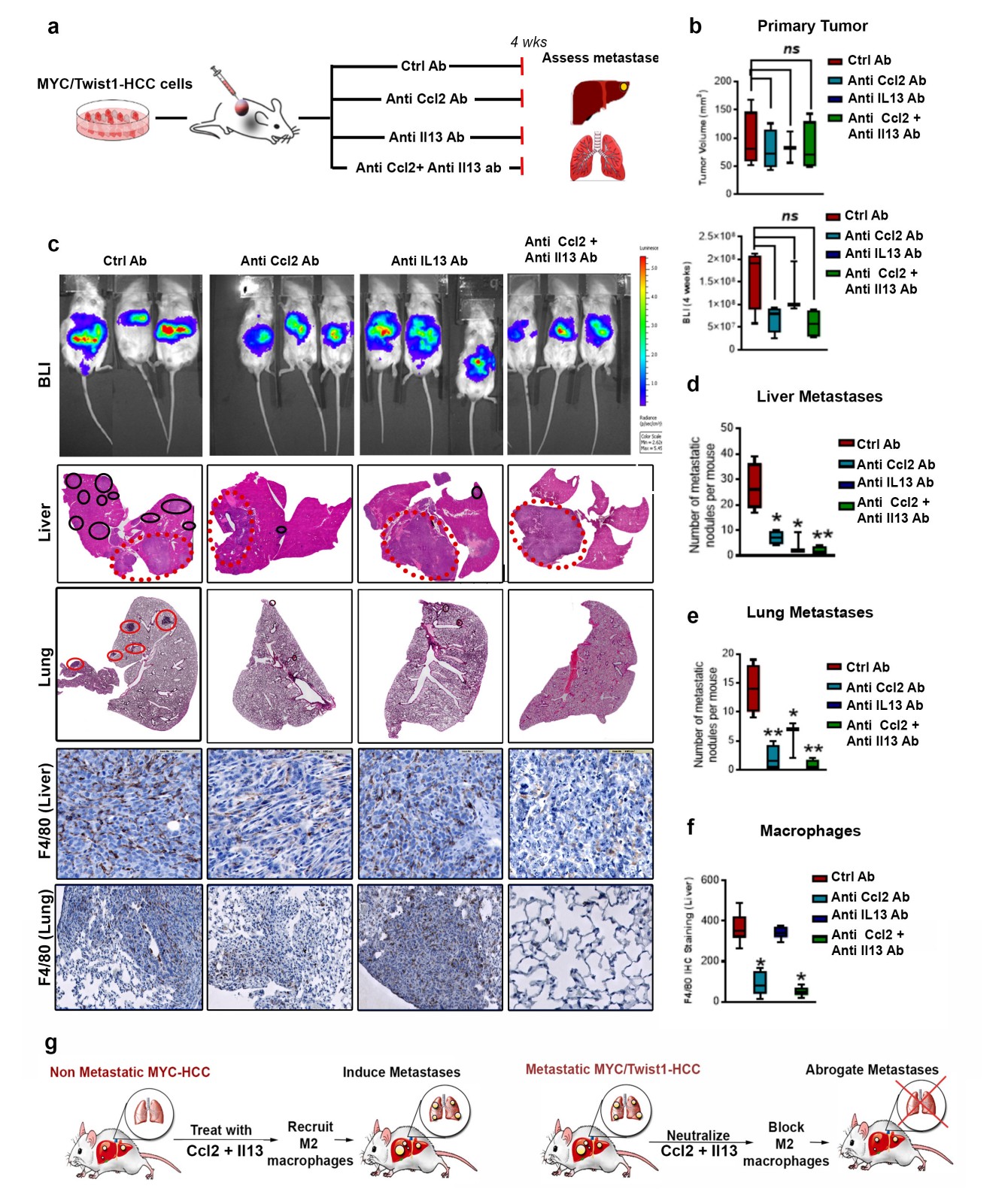

**Figure 7.** Combined inhibition of Ccl2 and Il13 has a synergistic effect on prevention of HCC metastasis. (**a**) Experimental scheme- Mice orthotopically transplanted with *MYC/Twist1*-HCC cells were treated either with control (ctrl) antibody or anti-Ccl2 antibody (ab) or anti-Il13 ab or anti-Ccl2+Il13 abs for 4 weeks. (**b**) Quantification of tumor volume of primary tumor and bioluminescence signal at 4 weeks in mice treated with ctrl ab (n = 4), anti-Ccl2 ab (n = 4) or anti-Il13 ab (n = 3) or anti-Ccl2+Il13 abs (n = 4). (**c**) BLI images, histopathology of liver (1X) and lungs (1X) of *MYC/Twist1*-HCC bearing mice

*Figure 7 continued*

treated with ctrl ab (n = 4) or anti-Ccl2 ab (n = 4) or anti-Il13 ab (n = 3) or anti-Ccl2+Il13 abs (n = 4) and IHC for F4/80 (20X) expression in the liver tumors and lungs in each group. The red dotted circles represent primary orthotopic tumor and the black circles show the metastatic nodules. (d) Quantification of number of metastatic nodules in the liver in *MYC/Twist1*-HCC bearing mice treated with ctrl ab or anti-Ccl2 ab or anti-Il13 ab or anti-Ccl2+Il13 abs. (**p<0.01). (e) Quantification of number of metastatic nodules in the liver in *MYC/Twist1*-HCC bearing mice treated with ctrl ab or anti-Ccl2 ab or anti-Il13 ab or anti-Ccl2+Il13 abs. (**p<0.01). (f) Quantification of macrophage infiltration in liver by IHC staining for F4/80 of *MYC/Twist1*-HCC bearing mice treated with ctrl ab (n = 4) or anti-Ccl2 ab (n = 4) or anti-Il13 ab (n = 3) or anti-Ccl2+Il13 abs (n = 4). (**p<0.01). (g) Schematic representation of how combined treatment of non-metastatic *MYC*-HCC can now induce metastases and synergistic inhibition of Ccl2 and Il13 can inhibit metastasis of *MYC/Twist1*-HCC.

The online version of this article includes the following figure supplement(s) for figure 7:

**Figure supplement 1.** Combined inhibition of Ccl2 and Il13 inhibits macrophage polarization.

Further, our results suggest that Ccl2 and Il13 alone can enable a non-metastatic cancer to become metastatic, cell non-autonomously. In this regard, it is notable that there are anecdotal reports suggesting that metastasis can occur during circumstances that would promote inflammation such as surgery (*Tohme et al., 2017*) or during infection (*Smith and Kang, 2013*).

Our findings have general relevance to human cancer. First, in 9502 patients with 33 different types of human cancer, the subset of tumors with *MYC* and *TWIST1* predicts poor survival, CCL2/IL13 expression and TAM infiltration. Second, in human patients with HCC, we found that CCL2, and IL13, were elevated in the plasma of patients with HCC and IL13 levels predicted invasive and aggressive HCC. Stratifying patients based on cytokine expression may help direct therapy. Third, the combined inhibition of CCL2 and IL13 profoundly impeded metastasis in our in vivo experimental model of liver cancer. Inhibition of CCL2 alone has not been effective in clinical trials for solid tumors (*Brana et al., 2015*; *Lim et al., 2016*; *Pienta et al., 2013*). We suggest that personalized therapy that combines the inhibition of CCL2 and IL13 is more likely to be effective.

Our work identifies two transcription factors, *MYC* and TWIST1, are key drivers of metastasis. Together, they elicit a cytokinome, that includes CCL2 and IL13, enabling crosstalk between cancer cells and host macrophages that drives tumor progression.

# Materials and methods

## Key resources table

| Reagent type (species) or resource | Designation | Source or reference | Identifiers | Additional information |
|---|---|---|---|---|
| Antibody | MYC (Rabbit, Monoclonal) | Epitomics | RRID:AB_11000313 | IHC (1:150), WB (1:1000) |
| Antibody | F4/80 (Rat, Monoclonal) | ThermoFisher | RRID: AB_10376289 | IHC (1:150) |
| Antibody | Twist1 (Mouse, Monoclonal) | Abcam | RRID:AB_883294 | WB (1:500) |
| Antibody | Phospho-Histone 3 (Rabbit, Polyclonal) | Cell Signaling Technology | RRID:AB_331535 | IHC (1:100) |
| Antibody | Cleaved Caspase3 (Rabbit, Polyclonal) | Cell Signaling Technology | RRID:AB_2341188 | IHC (1:100) |
| Antibody | Neutrophil (Rat, Monoclonal) | Abcam | RRID:AB_881409 | IHC (1:100) |
| Antibody | CD4 (Mouse, Monoclonal) | Abcam | RRID:AB_2686917 | IHC (1:1000) |
| Antibody | Glutamine Synthetase (Rabbit, Monoclonal) | Abcam | RRID:AB_446132 | IHC (1:200) |

*Continued on next page*

*Continued*

| Reagent type (species) or resource | Designation | Source or reference | Identifiers | Additional information |
|---|---|---|---|---|
| Antibody | CD31 (Rabbit, Polyclonal) | Abcam | RRID:AB_726362 | IHC(1:100) |
| Antibody | CD206 (Mouse, Monoclonal) | R and D | RRID:AB_2745540 | ICC (1:100) |
| Antibody | CCL2 (Mouse, Monoclonal) | BioXCell | RRID:AB_10950302 | In vivo treatment (10 mg/kg body weight three times per week) |
| Antibody | Il-4 (Mouse, Monoclonal) | Genentech | Propreitary | In vivo treatment (10 mg/kg body weight three times per week) |
| Antibody | IL-13 (Mouse, Monoclonal) | Genentech | Propreitary | In vivo treatment (10 mg/kg body weight three times per week) |
| Genetic Reagent (*M. musculus*) | *Twist1-tetO7-luc* | PMID: 22654667 | Twist1 Transgenic mouse model | Felsher Lab |
| Genetic Reagent (*M. musculus*) | *LAP-tTA* | PMID: 15475948 | LAP-tTA mouse | Felsher Lab |
| Genetic Reagent (*M. musculus*) | *TetO-MYC* | PMID: 10488335 | MYC Transgenic mouse | Felsher Lab |
| Genetic Reagent (*M. musculus*) | Nod Scid-Gamma (NSG) mice | Jackson Laboratory | MGI:3577020 | Felsher Lab |
| Cytokine | CCL2 | Peprotech | RRID:AB_147738 | In vivo treatment (500 ng/mouse) |
| Cytokine | IL13 | Peprotech | RRID:AB_147840 | In vivo treatment (250 ng/mouse) |
| Cytokine | IL4 | Peprotech | RRID:AB_147635 | In vivo treatment (250 ng/mouse) |
| Drug | Clodronate liposomes | Liposoma | CP-005–005 | In vivo treatment (6.5 µl/g body weight/mouse) |
| Sequence-based reagent | MYC | Stanford PAN facility | N/A | f-CTGCGACGAGGAGGAGAACT r-GGCAGCAGCTCGAATTTCTT |
| Sequence-based reagent | UBC | Stanford PAN facility | N/A | f-AGCCCAGTGTTACCACCAAG r-ACCCAAGAACAAGCACAAGG |
| Sequence-based reagent | Twist1 | Stanford PAN facility | N/A | f-GGACAAGCTGAGCAAGATTCA r-CGGAGAAGGCGTAGCTGAG |
| Sequence-based reagent | CD206 | Stanford PAN facility | N/A | f-CTCTGTTCAGCTATTGGACGC r-TGGCACTCCCAAACATAATTTGA |
| Sequence-based reagent | Arginase-1 | Stanford PAN facility | N/A | f-CTCCAAGCCAAAGTCCTTAGAG r-AGGAGCTGTCATTAGGGACATC |
| Sequence-based reagent | CCR2 | Stanford PAN facility | N/A | f-GTGTACATAGCAACAAGCCTCAAAG r-CCCCCACATAGGGATCATGA |
| Sequence-based reagent | INFaR | Stanford PAN facility | N/A | f-CTTCCACAGGATCACTGTGTACCT r-TTCTGCTCTGACCACCTCCC |
| Sequence-based reagent | iNOS | Stanford PAN facility | N/A | f-ACACCGACCCGTCCACAGTAT r-CAGAGGGGTAGGCTTGTCTC |
| Sequence-based reagent | CX3CR1 | Stanford PAN facility | N/A | f-TACCTTGAGGTTAGTGAACGTCA r-CGCTCTCGTTTTCCCCATAATC |

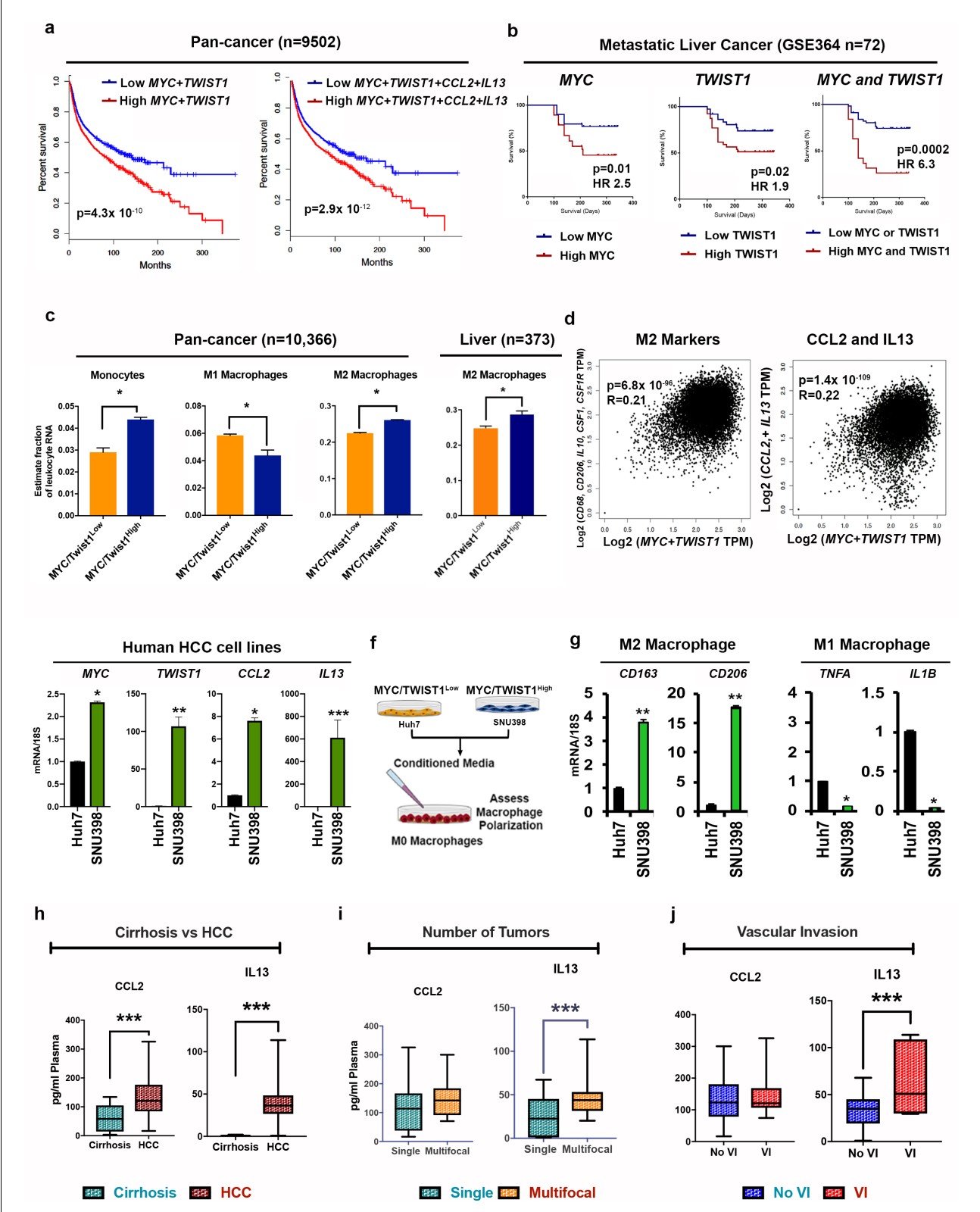

**Figure 8.** MYC and TWIST1 predict poor prognosis, TAM infiltration and pro-TAM cytokines in 33 human cancers. (**a**) Disease-free survival in pan-cancer TCGA cohort of 9502 patients from 33 cancers stratified by median *MYC+TWIST1* expression (first KM curve) and median *MYC+TWIST1+* CCL2+IL13 (second KM curve). (**b**) Disease-free survival in GSE364 cohort of metastatic HCC patients (n = 144) stratified by median *MYC* and *TWIST1* expression. (**c**) Comparison of relative percentage of monocyte and macrophage subpopulations, derived using CIBERSORT analysis of pan cancer TCGA cohort of

*Figure 8 continued on next page*

*Figure 8 continued*

10366 patients from 33 cancers with tumors stratified as *MYC*/*TWIST1*[High] or *MYC*/*TWIST1*[low] (*p<0.05). Comparison of relative percentage of M2 macrophage subpopulation, derived using CIBERSORT analysis of Liver cancer TCGA cohort of 373 patients with tumors stratified as *MYC*/*TWIST1*[High] or *MYC*/*TWIST1*[low] (*p<0.05). (d) Correlation of *MYC+TWIST1* expression with TAM markers in TCGA pan-cancer cohort and HCC cohort. Correlation of *MYC+TWIST1* expression with CCL2 and IL13 in TCGA pan-cancer cohort (n = 9502). (e) Comparison of mRNA expression of *MYC*, *TWIST1*, *CCL2* and *IL13* between two HCC cell lines Huh7 (n = 3 biological replicates with three technical replicates each) and SNU398 (n = 3 biological replicates with three technical replicates each). (f) Experimental scheme for analysis of macrophage polarization in vitro assay. (g) Expression of M2 (CD163, CD206) and M1 markers (TNFA, IL1B) after treatment with conditioned media from Huh7 (n = 3 biological replicates with three technical replicates each) or SNU398 (n = 3 biological replicates with three technical replicates each). (*p<0.05, **p<0.01, ***p<0.001). (h) Prospective clinical study of plasma levels of cytokines in patients with cirrhosis (n = 10) versus patients with HCC (n = 25). (*p<0.05, **p<0.01, ***p<0.001). (i) Comparison of plasma levels of cytokines between patients with single HCC versus multifocal HCC (*p<0.05, **p<0.01, ***p<0.001). (j) Comparison of plasma levels of cytokines between patients with HCC with vascular invasion and without vascular invasion (*p<0.05, **p<0.01, ***p<0.001).

The online version of this article includes the following figure supplement(s) for figure 8:

**Figure supplement 1.** M2 macrophages have a prognostic role in human cancers including liver cancer.

## Transgenic mice

Mouse *Twist1* cDNA was PCR cloned into the bidirectional *tetO7* vector S2f-*IMCg* at *EcoR*I and *Not*I sites, replacing the *eGFP* ORF. The resultant construct, *Twist1-tetO7-luc* (*Tran et al., 2012*), was sequenced, digested with *Kpn*I and *Xmn*I, and used for injection of FVB/N pronuclei by the Stanford

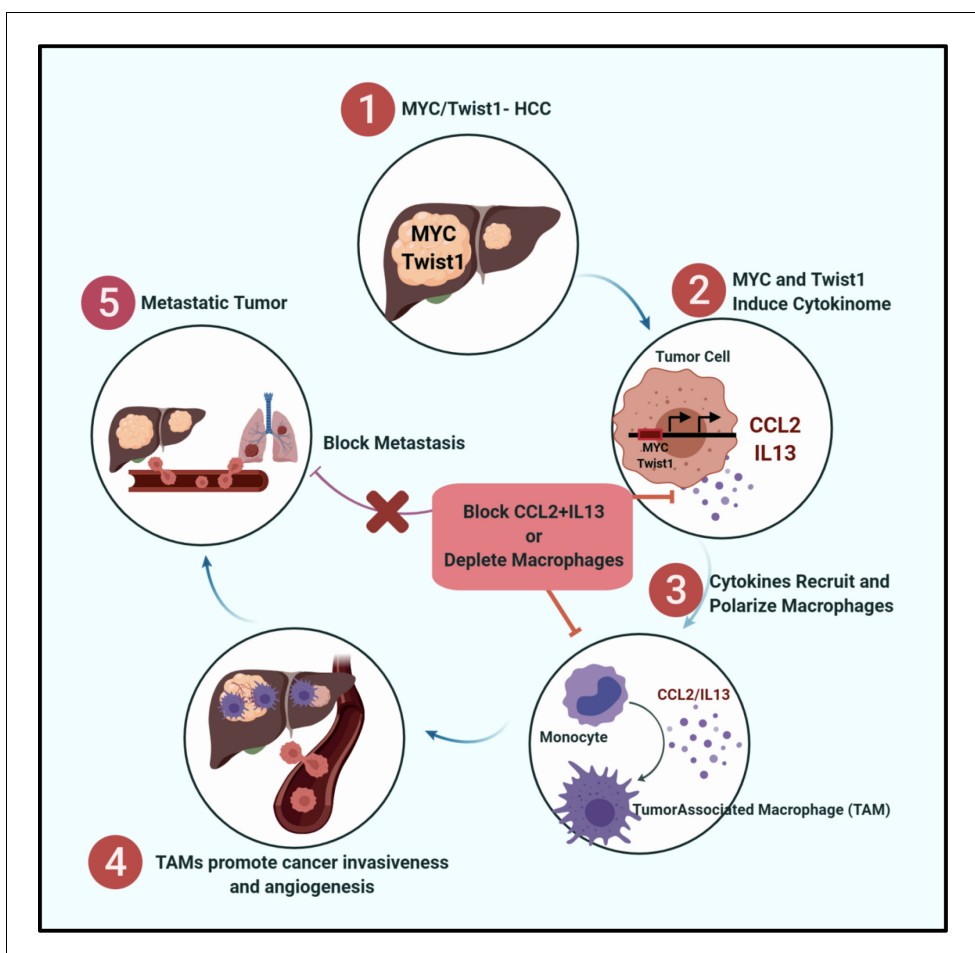

**Figure 9.** MYC and Twist1 Drive Metastasis via CCL2/IL13 mediated macrophage activation. Graphical representation showing that MYC and Twist1 elicit a cytokinome including CCL2/IL13 which mediate the crosstalk between cancer cells and macrophages. Therapeutic blockade of the cytokines can inhibit cancer metastasis.

Transgenic Facility. Founders were screened by genotyping using PCR. Founders were mated to LAP-tTA mice, and BLI was used to additionally screen for functional *Twist1-tetO7-luc* founders, subsequently termed LAP-tTA/TRE-*Twist1*/Luc. The LAP-*tTA, and TetO-MYC* transgenic lines have been described previously (*Felsher and Bishop, 1999*; *Kistner et al., 1996*; *Shachaf et al., 2004*). LAP-tTA/TRE-*Twist1*/Luc mice were mated to LAP-tTA/TRE-*MYC* mice, and progeny were screened by PCR. The final background of the mouse was FVB/N. Doxycycline (Dox- Sigma) was administered in the drinking water weekly at 0.1 mg/mL during mating and continuing until mice reached 6 weeks of age. Animals were euthanized upon disease morbidity as assessed by tumor burden. Macrometastases were assessed upon necropsy and tissues were collected and stored for further analysis. All procedures were performed in accordance with APLAC protocols and animals were housed in a pathogen-free environment.

## Small animal imaging

in vivo bioluminescent imaging (BLI) was utilized to confirm oncogene activation in transgenic mice beginning one week before, and continuing each week following, Dox removal. BLI was performed on an IVIS Spectrum (Caliper Life Sciences, Hopkinton, MA). Briefly, mice were injected i.p. with the substrate D-Luciferin (150 mg/kg) and then anesthetized with 2.5% isoflurane delivered by the Xenogen XGI-8 5-port Gas Anesthesia System. Animals were then placed into the IVIS Spectrum, and Living Image Software was used to collect, archive, and analyze photon fluxes and transform them into pseudocolor images.

MRI scans were performed using a 7T small animal MRI scanner (Bruker Inc, Billerica, MA, Stanford Small Animal Imaging Facility, CA) equipped with a 40 mm Millipede RF coil (ExtendMR LLC, Milpitas, CA). Under anesthesia by inhalation of 1–3% isoflurane mixed in with medical-grade oxygen *via* nose-cone, and acquisitions were gated using the respiratory triggering. For tumor detection, a respiration triggered T2-weighted 3D turbo spin echo sequence was used (TR/TE 3000/205 ms, voxel size (0.22 mm$^3$). The isotropic voxel size of 0.22 mm in all directions provides a high in plane and across plane resolution. Thereby, the location of one tumor could be defined in all three orientations using specific landmarks, such as major vessels or other tumors. T2-weighted anatomical imaging was performed approximately once weekly. Anatomical and parametric images were analyzed and tumor volumes were measured using Osirix image processing software (Osirix, UCLA, and Los Angeles, CA).

## Cell culture

Conditional HCC cell lines were derived from LAP-tTA and TetO-*MYC* or -*MYC*/*Twist1* mice. Cells were grown in DMEM (Invitrogen), supplemented with 10% FBS (Invitrogen), and cultured at 37˚C in a humidified incubator with 5% CO2. Cell lines were confirmed to be negative for Mycoplasma contamination.

## Orthotopic transplantation assay

An orthotopic mouse model was established by transplanting mouse *MYC-* or *MYC*/*Twist1*-HCC tumors (1 mm$^3$) under the liver capsule of NOD/Scid/Gamma (NSG) recipient mice. Bioluminescent (BLI) and MRI scans monitoring are used to monitor tumor engraftment and growth. Mice are euthanized once predetermined specific endpoints are met or based on morbidity whichever occurs first.

## Intravenous transplantation assay

An intravenous transplantation mouse model was established by tail vein intravenous injection of 500,000 *MYC-* or *MYC*/*Twist1*-HCC cells of NOD/Scid/Gamma (NSG) recipient mice. Bioluminescent (BLI) monitoring is used to monitor tumor metastasis in lungs. Mice are euthanized once predetermined specific endpoints are met or based on morbidity whichever occurs first.

## RNA sequencing

RNA sequencing of *MYC*-HCC and *MYC*/*Twist1*-HCC was performed at the Beijing Genomics Institute (BGI) using their BGIseq 500 platform single end 150 bp, 20 million reads per sample. Genes expression level is quantified by a software package called RSEM. We counted the number of identified expressed genes and calculated its proportion to total gene number in database for each

sample RNA sequencing data are deposited in Gene Expression Omnibus (GEO). DEseq software was used to perform differential expression analysis. Ingenuity Pathway Analysis (IPA, Qiagen) was used to perform functional pathway analysis.

Principal-component analysis (PCA; Qlucore Omics Explorer v. 2.2) was used to generate a visually interpretable overview of the transcriptional profile of *MYC*-HCC and *MYC/Twist1* HCC. Qlucore Omics Explorer was used to structure data to verify if tumor subgroups could be identified. Variance filtering was used to reduce the noise, and the projection score to set the filtering threshold. The 'mean = 0, var = 1' setting was used to scale the data. PCA was used to visualize the data set in a three-dimensional space, after filtering out variables with low overall variance to reduce the impact of noise, and centering and scaling the remaining variables to zero mean and unit variance. The projection score was used to determine the optimal filtering threshold, retaining N variables (*Soneson and Fontes, 2011*).

## In vivo treatment mouse models

For antibody treatment, mice were injected i.p. with isotype control IgG or anti-Ccl2 (BioXcell), -Il13 and –Il4 (Genentech) antibody (10 mg/kg body weight three times per week). For recombinant cytokines treatment, mice were injected i.p. three times per week with PBS or Ccl2 (Peprotech, 500 ng/mouse), Il4 and IL4 (Peprotech, 250 ng/mouse). For clodronate liposomes (CL) treatment, CL or control liposomes were administered i.p. at 6.5 µl/g body weight 3 times per week to NSGs mice. CL and control treatments were administered to NSG mice previously injected i.v. with $0.5 \times 10^6$ *MYC/Twist1*-HCC cells or orthotopically transplanted with *MYC/Twist1*-HCC tumors. Experimental and control mice were killed 4 weeks after tumor were transplanted. Primary tumors and lung metastases were collected for H and E staining and IHC.

## Immunohistochemistry and immunofluorescence

Paraffin embedded tumor sections were deparaffinized by successive incubations in xylene, graded washes in ethanol, and deionized water. Epitope unmasking was performed by steaming in DAKO antigen retrieval solution for 45 min. Paraffin embedded sections were immunostained with *MYC* (1:150, Epitomics), or cleaved caspase 3 (1:100, Cell Signaling technology), phospho histone 3 (1:200, Cell Signaling Technology), F4/80 (1:50, ThermoFisher), Cd4 (1:1000, Abcam), Neutrophil (1:100, Abcam), overnight at 4°C. The tissue was washed with PBS and incubated with biotinylated anti-rabbit, anti-rat or anti-mouse for 30 min at room temperature (1:300 Vectastain ABC kit, Vector Labs). Sections were developed using 3,3'- Diaminobenzidine (DAB, Vector Labs), counterstained with hematoxylin, and mounted with Permount. Images were obtained on a Philips Ultrafast Scanner. All IF and IHC experiments were conducted using at least three biological replicates per group.

## Quantification of IHC staining analysis

Images are analyzed in Icy (BioImage Analysis Unit, Paris, http://icy.bioimageanalysis.com). Areas of immunopositivity are selected for positive values in the Color Picker threshold tool in the support vector machine (SVM) tab. Unstained nuclei and cytoplasm areas are chosen for negative values. Default values are used for the kernel. Immunopositive areas are selected as regions of interest (ROIs). Subsequently, ROIs are separated. Those with interior size <68 pixels are eliminated from the ROI table as they correspond to small specks of non-specific immunopositivity, whereas larger areas corresponded closely to distinct positive cells. Counts were obtained from the ROI tables.

## Wound healing assay

$1 \times 10^6$ *MYC*- or *MYC/Twist1*-HCC cells per well were seeded in 6-well plate and cultured overnight in culture medium. Thereafter, a scratch (wound) was introduced in the confluent cell layer using a yellow tip. Cells were washed three times to remove detached cells. Cells were then incubated with supernatant of primary macrophages harvested from *MYC*- or *MYC/Twist1*-HCC primary tumors for 24 hr. Pictures of a defined wound spot were made with a Leica DM16000 microscope at t = 0, 24, 48 and 72 hr. The area of the wound in the microscopic pictures was measured using Image J software (National Institutes of Health, MD). The percentage wound healing after 72 hr was calculated in relative to the total wound area at t = 0 hr of the same wound spot. To isolate tumor associated macrophages from liver tumors, we first cut the tumor into small pieces, digested with collagenase

and incubated in a culture dish at 37°C for 1–2 hr. We removed and discarded all detached cells and/or dead cells at 2 hr. Next, we maintained the remaining cells in culture for 24 hr and then removed any adherent tumor cells by fast trypsinization. Macrophages are highly adherent and are the main population that remains adherent beyond this step. We harvested the macrophages and confirmed their phenotype and polarization states by qPCR (*Figure 2f*).

## Transwell chamber migration assay

Cell migration was assessed using a 12 well transwell chamber with 8 µm filter inserts (Corning). Raw macrophages (264.7 macrophage cell line (ATCC)) were seeded in the upper chamber and *MYC/Twist1* cells were seeded in the lower chamber. Neutralizing antibody to Ccl2, Ccl5, Ccl7, Cxcl1, Il13 or Il4 were added to the lower chamber in triplicates. After 16 hr the migrated cells were fixed in 4% paraformaldehyde (PFA) and 100% methanol. The non-migrated cells were gently removed with a swab. Cells in the lower surface of the membrane were stained with 0.5% crystal violet for 20 mins. The membranes were imaged and number of macrophages in 10 random fields were counted. The experiment was performed in triplicates.

## 3d co-culture system of macrophages and HCC cells

The 24-well plates were precoated with $10 \times 10^3$ mouse Raw 264.7 macrophages (ATCC) resuspended in 200 ul of Matrigel growth factor reduced (Corning) for 30 min at 37°C. Then, $80 \times 10^3$ *MYC* or *MYC/Twist1*-HCC cells resuspended in 200 ul of Matrigel and directly seeded onto the 24-well plate precoated with matrigel+macrophage mixture. The cells were incubated at 37°C for up to 1 week to allow the spheroids to form. Recombinant Ccl2 (50 ng/ml) or Il13, IL4 (25 ng/ml each) and anti-Ccl2 antibody (30 ug/ml), or anti–Il13 antibody, anti-Il4 antibody (20 µg/ml each) were added directly to the coculture and refreshed every 48 hr. Spheroids were fixed with 10% PFA overnight, paraffin embedded and sectioned (4–5 µm) as previously described (*Ootani et al., 2009*). Sections were deparaffinized and stained with H and E for the initial histology analysis. For further immunohistochemistry analysis, we used F4/80 antibody as described above. All assays were performed at least 3 times.

## Quantitative Real-Time PCR

RNA was isolated using RNeasy plus mini kit according to the manufacturer's instructions (Qiagen). cDNA was synthesized using SuperScript III (ThermoFisher). qPCR was performed using specific primers (Key Resources) and SYBR Green (Roche) in an Applied Biosystems Real Time PCR System (Life Technologies). Data were normalized to *UBC*. A minimum of 3 biological and three technical replicates were used for all qPCR experiments.

## Luminex – eBioscience/Affymetrix Magnetic bead Kits

This assay was performed in the Human Immune Monitoring Center at Stanford University. Human 62-plex or Mouse 38 plex kits were purchased from eBiosciences/Affymetrix and used according to the manufacturer's recommendations with modifications as described below. Briefly: Beads were added to a 96 well plate and washed in a Biotek ELx405 washer. Samples were added to the plate containing the mixed antibody-linked beads and incubated at room temperature for 1 hr followed by overnight incubation at 4°C with shaking. Cold and Room temperature incubation steps were performed on an orbital shaker at 500–600 rpm. Following the overnight incubation plates were washed in a Biotek ELx405 washer and then biotinylated detection antibody added for 75 min at room temperature with shaking. Plate was washed as above and streptavidin-PE was added. After incubation for 30 min at room temperature wash was performed as above and reading buffer was added to the wells. Each sample was measured in duplicate. Plates were read using a Luminex 200 instrument with a lower bound of 50 beads per sample per cytokine. Custom assay Control beads by Radix Biosolutions are added to all wells.

## Immune cell deconvolution analysis

The CIBERSORT gene expression deconvolution package was used to estimate the immune cell composition in the *MYC*- and *MYC/Twist1*-HCC. The LM22 signature was used as the immune cell gene signature. We modified it for studying mouse immune subsets by carefully converting the

genes in the signature to their respective mouse orthologs. The settings for the run were: 1000 permutations with quantile normalisation disabled. The student T-test was used to infer the statistical significance of the predicted immune cell populations where p<0.05 was considered significant.

## TCGA analysis

The pan cancer RNAseq data was downloaded from the GDC data portal https://portal.gdc.cancer.gov/ on Oct 15, 2018. Spearman test was used for correlation analysis. Kaplan Meier analysis was performed for survival analysis. K means clustering was used to stratify patients into two groups based on *MYC* and *TWIST1* expression.

## Clinical study

This study was approved by the institutional review board (IRB) of Stanford University (IRB Number: 28374), and all patients provided informed consent before being enrolled in this study. We prospectively collected blood from patients with HCC or cirrhosis alone. Blood samples were obtained at the time of diagnosis of HCC. Plasma was separated, aliquoted and stored at −80C. Luminex assay to measure cytokine levels was performed as mentioned above. Clinical data was gathered from their medical records.

## Statistics

Differences between groups were analyzed using Student's t-test or one-way analysis of variance (ANOVA). Chi square test was used to compare categorical variables. Kaplan Meier analysis with Log Rank test was performed for survival analysis. A *P* value of less than 0.05 was considered to be significant and is indicated by one asterisk (*), a *P* value of less than 0.01 is indicated by two asterisks (**), a *P* value of less than 0.001 is indicated by three asterisks (***), and a *P* value of less than 0.0001 is indicated by four asterisks (****). All graphs are presented as the mean + /- SEM. Analyses were performed with Prism, version 5 (GraphPad Software, San Diego, CA).

## Acknowledgements

RD- National Institutes of Health (NIH) grant CA222676 from the National Cancer Institute (NCI), American College of Gastroenterology Junior Faculty Career Development Grant. PTT- Ronald Rose and Joan Lazar; Movember Foundation, Prostate Cancer Foundation; NIH/NCI (R01CA166348, U01CA212007, U01CA231776 and 1R21CA223403). VB- FRM- Fondation pour la Recherche Médicale grant Joseph R Arron, Genentech, USA- for providing the control antibody, the anti-Il4 and the anti-Il13 antibodies. DF- National Institutes of Health (NIH) grant CA208735 from the National Cancer Institute (NCI). Mindie Nguyen- provided plasma samples from control patients with cirrhosis. Pauline Chu- Helped with mouse histology services.

## Additional information

### Funding

| Funder | Grant reference number | Author |
| --- | --- | --- |
| National Cancer Institute | CA208735 | Dean W Felsher |
| National Cancer Institute | CA222676 | Renumathy Dhanasekaran |
| National Cancer Institute | R01CA166348 | Phuoc T Tran |
| National Cancer Institute | U01CA212007 | Phuoc T Tran |
| National Cancer Institute | U01CA231776 | Phuoc T Tran |
| National Cancer Institute | 1R21CA223403 | Phuoc T Tran |
| Fondation pour la Recherche Médicale | FRM grant | Virginie Baylot |
| American College of Gastroenterology | ACG Career Development Grant | Renumathy Dhanasekaran |

The funders had no role in study design, data collection and interpretation, or the decision to submit the work for publication.

## Author contributions
Renumathy Dhanasekaran, Conceptualization, Resources, Data curation, Formal analysis, Validation, Investigation, Visualization, Methodology, Writing—original draft, Writing—review and editing; Virginie Baylot, Conceptualization, Data curation, Formal analysis, Validation, Investigation, Visualization, Methodology, Writing—review and editing; Minsoon Kim, Validation, Investigation, Visualization, Methodology; Sibu Kuruvilla, Validation, Investigation, Methodology; David I Bellovin, Conceptualization, Investigation, Methodology; Nia Adeniji, Ian Lai, Meital Gabay, Ling Tong, Maya Krishnan, Investigation, Methodology; Anand Rajan KD, Formal analysis, Investigation, Methodology; Jangho Park, Data curation, Investigation; Theodore Hu, Data curation, Formal analysis, Investigation; Mustafa A Barbhuiya, Resources, Data curation, Formal analysis, Investigation; Andrew J Gentles, Data curation, Formal analysis; Kasthuri Kannan, Formal analysis, Investigation, Visualization; Phuoc T Tran, Dean W Felsher, Conceptualization, Resources, Supervision, Funding acquisition, Project administration, Writing—review and editing

## Author ORCIDs
Renumathy Dhanasekaran (iD) https://orcid.org/0000-0001-8819-7511
Virginie Baylot (iD) https://orcid.org/0000-0003-1313-2857
Nia Adeniji (iD) https://orcid.org/0000-0002-7957-5987
Ian Lai (iD) https://orcid.org/0000-0001-8662-7290
Ling Tong (iD) https://orcid.org/0000-0003-4634-9462
Jangho Park (iD) https://orcid.org/0000-0002-6151-5193
Mustafa A Barbhuiya (iD) https://orcid.org/0000-0002-8046-5064
Phuoc T Tran (iD) https://orcid.org/0000-0002-0147-0376
Dean W Felsher (iD) https://orcid.org/0000-0003-2496-523X

## Ethics
Human subjects: This study was approved by the institutional review board (IRB) of Stanford University (IRB Number: 28374), and all patients provided informed consent before being enrolled in this study.
Animal experimentation: All procedures were performed in strict accordance with APLAC protocols (#14045) and animals were housed in a pathogen-free environment. All invasive procedures and surgery were performed under appropriate anesthesia and analgesia, and every effort was made to minimize suffering.

## Decision letter and Author response
Decision letter https://doi.org/10.7554/eLife.50731.sa1
Author response https://doi.org/10.7554/eLife.50731.sa2

# Additional files

## Supplementary files
• Supplementary file 1. List of genes (220 up and 294 down) that were differentially expressed between the *MYC*-HCC and *MYC/Twist1*-HCC.

• Supplementary file 2. The top biological processes, and associated genes, upregulated in *MYC/Twist1*-HCC versus *MYC*-HCC.

• Transparent reporting form

## Data availability
All data generated or analyzed during this study are included in the manuscript and supporting files. Data has been deposited in GEO under accession codes GSE135878.

The following dataset was generated:

| Author(s) | Year | Dataset title | Dataset URL | Database and Identifier |
|-----------|------|---------------|-------------|-------------------------|
| Dhanasekaran R, Felsher D | 2019 | MYC and Twist1 Expression in Cancer Cells Activates Host Macrophages to Enable metastasis | https://www.ncbi.nlm.nih.gov/geo/query/acc.cgi?acc=GSE135878 | NCBI Gene Expression Omnibus, GSE135878 |

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
