## [Decision Letter]

Thank you for submitting your article "MYC and *Twist1* expression in cancer causes macrophage activation that enables metastasis" for consideration by *eLife*. Your article has been reviewed by three peer reviewers, and the evaluation has been overseen by a Reviewing Editor and Jeffrey Settleman as the Senior Editor. The following individuals involved in review of your submission have agreed to reveal their identity: Jung-Eun Lee (Reviewer #1).

The reviewers have discussed the reviews with one another and the Reviewing Editor has drafted this decision to help you prepare a revised submission.

Summary:

The reviewers' comments can be addressed without performing extensive new experiments. Most of the points can be ironed out by textual changes or data re-analysis. In this spirit, major point 5 raised by reviewer 3 does not have to be addressed experimentally. It is sufficient to discuss this point.

Reviewer #1:

Summary:

1) The authors generated a transgenic mouse model whereby conditional expression of MYC and *Twist1*1 promotes the metastasis of hepatocellular carcinoma (HCC) in most mice. Metastatic tumors are reversible upon inactivation of *Twist1*.

2) They find that tumor-associated macrophages (TAM) are recruited to HCC in this mouse model.

3) Treatment with CCL2 and IL13 induced MYC-HCCs to metastasize whereas blockade of CCL2 and IL13 or depletion of macrophages abrogated MYC/*Twist1*-HCC metastasis.

4) High expression of MYC and *Twist1* predicts poor survival of cancer patients.

5) The plasma level of IL13 correlates with the vascular invasion in HCC.

General assessment:

To establish a model that enables the metastasis of murine MYC-driven HCC, the authors introduced *Twist1*, which is known to promote the metastasis of cancer cells, to their previously reported murine MYC-driven HCC model (Tet-off conditional expression system).

Here they present a mechanism by which MYC and *Twist1* drive the metastasis of HCC. This study represents technically solid work. It is striking that *Twist1* expression increases the probability of the metastasis of MYC-driven HCC by 45 folds.

However, the conceptual novelty of the suggested mechanism is somewhat limited by a previous publication (Cancer Res. 2013. *Twist1* induces CCL2 and recruits macrophages to promote angiogenesis) that reported the same role of *Twist1* for recruiting macrophages through CCL2 and promoting angiogenesis around carcinoma cells. Given that angiogenesis is required for metastasis, this study confirms the same role of CCL2 in the metastasis of liver cancer while reporting a new role of IL13 in M2 polarization in this process. The authors should discuss this reference in more detail to highlight the novelty of this study.

An interesting twist of this study is that *Twist1* somehow needs MYC to promote the metastasis of liver cancer. Given that MYC and *Twist1* have been reported to cooperate, if the authors can shed some light on the mechanism underlying this dependency, it will increase the impact of this study. Does MYC increase the transcriptional factor activity of *Twist1*, for example? Is IL13-mediated M2 polarization required for angiogenesis?

Regardless of this gap of knowledge, this study will be of interest to the liver cancer community now that the authors show that the combined inhibition of CCL2 and IL13 suppress the metastasis in a mouse model of liver cancer and that the plasma level of IL13 correlates with the vascular invasion in HCC.

Reviewer #2:

This paper first describes a model of liver cancer, induced by *MYC+TWIST1* genes. The authors report the important discovery that tumors arising from the combination of the two genes induces metastasis, whereas only *MYC* or *TWIST1* does not. The work shows that the *MYC+TWIST1* tumors express the cytokines CCL2 and IL13, which polarize tumor-associated macrophages of the type M2. Moreover, systemic administration of CCL2 and IL13 was sufficient to induce metastasis. Although it is known that the tumor-associated macrophages promote tumor and metastasis development, the mechanism proposed to explain the higher metastatic potential of the *MYC+TWIST1* tumors is novel and interesting.

Consistently, analysis of different human cancers shows that *MYC* and *Twist1* are strongly associated with poor overall survival. In a prospective study, they show that higher plasmatic levels of CCL2 and IL13 are associated with metastasis. Finally, they also show that metastasis in this model are inhibited by Ccl2+Il13 neutralizing antibodies, which gives clues for anti-metastasis approaches in oncology.

Altogether, the work shows relevant and novel information, and the technological approach is sound. It is novel and interesting that *MYC* and *Twist1* induce metastasis by modulating the immune microenvironment and not by inducing proliferation or invasion. However, the work does not elucidate whether *MYC* activate the transcription of the CCL2 and/or IL13 genes or a different mechanism is in place

1) The work shows that *MYC* induces the expression of CCL2 and IL13 and that these cytokines are sufficient for the *MYC*-induced liver tumors to metastasize. This is a relevant information. However, whether IL13 or CCL2 are direct transcriptional targets of *MYC+TWIST1* remains to be elucidated. The authors may at least ask in the databases whether the DNA region around the transcription start site of IL13 and CCL2 genes contain *MYC* and *Twist1* binding motifs and to show *MYC* and *Twist1* binding by ChIP. Actually the binding of *MYC* to the genes can be also tested I many cell lines from the public data of the ENCODE consortium.

2) Figure 1C: It is difficult to tell from these images whether the luciferase is being expressed in the liver of the animal, or another organ of the body or in the peritoneum. Maybe they can use another technique to show *Twist1* in the liver upon Dox withdrawal. For example, and IHQ or IF for Twist or Luc, a western blot or even RT-PCR showing the *Twist1* RNA expression).

Also, they should give information on the penetrance of the *MYC-Twist1*/Luc – What percentage of mice show it? At the same intensity? It does not look so from the two mice of the figure.

3) Figure 1—figure supplement 1B: As the authors rightly say, it is important to rule out an effect of *Twist1* on *MYC* levels. However, the figure apparently shows an increase in *MYC* levels in the *Twist1-MYC* livers, if the reduced levels of Actin are considered (compare lanes 1 and 3). A better western or a quantification of this blot normalizing against actin would be required to make the conclusion.

4) Subsection “*MYC* and *Twist1* cooperate to remodel the tumor immune microenvironment” paragraph two, Figure 2—figure supplement 1E: "the inactivation of *MYC* and *Twist1* in tumors shows rapid tumor regression". However, no quantification is given on the extent of tumor repression upon the addition of Dox to the water. In how many mice? Do the tumors completely regress? How long before they regress?

5) Figure 4C: It is known that the macrophage subtype M2a is the one activated by IL13. So it will be interesting to check whether this is the subtype polarized in the *MYC*+*TWIST1* liver tumors.

6) Subsection “Combined inhibition of CCL2 and IL13 synergistically inhibits metastasis in vivo”: The liver tumor burden in response to the neutralizing antibodies must be quantified (e.g. by densitometry of the images).

Reviewer #3:

General assessment:

The manuscript by Baylot et al. shows a new metastastic mouse model of HCC, based on the cooperative function of *MYC* and *Twist1*. The authors provide evidence that *MYC* and *Twist1* together are responsible for the induction of a cytokinome, including CCL2 and IL13, which cause macrophage tumor infiltration and their transition to a metastasis-promoting M2 phenotype. The story is very interesting by itself, but it is probably just the beginning of further studies needed to define the distinct contributions of *MYC* and *Twist1* to the metastatic phenotype (unfortunately *MYC* and *Twist1* are both under the same Tet System control and cannot be turned off separately in the *MYC/Twist1* mice). Also, there is no mechanism explaining exactly how the macrophages are promoting metastasis, even though it is clear that they are needed for the process.

Summary of substantive concerns:

1) General comment for figures: the scale bars are missing and sometimes the same figure includes the comparison among panels with different magnifications (i.e. Figure 1J-K, lung and liver panels, or Figure 2G-H). That needs to be corrected in all figures.

2) Regarding the same Figures 1J-K, in subsection “*Twist1* induces spontaneous metastatic progression of *MYC*-driven HCC in vivo” the authors state that *MYC* levels are similar in the two tumor models, but the images do not seem to agree with that statement: in *MYC/Twist1* HCC top left panel, *MYC* seems much higher that the corresponding *MYC* HCC panel, while in the merge it looks exactly the opposite. The authors should quantify the images in order to make conclusions and correct for different exposure times.

3) The authors state that "Macrophage depletion with clodronate liposomes […] had reduced intrahepatic and lung metastases". However, the figure clearly shows a difference also in the primary tumor burden, which could in part account for the reduction in mets. This is definitely a point that needs to be clarified.

4) The authors should at least comment on the fact that *Twist1* is a bHLH protein that can bind the same Ebox as *MYC* does, hence not only possibly cooperating, but even competing with it. Again, some characterization of this molecular aspects of target regulation might be relevant to future studies.

5) Orthotopic and intravenous transplantation assays: To perform this in vivo assay the authors state that the tumors or cells are inoculated into NSG mice. Why do the authors perform these experiments with an immunodeficient mouse strain? Why don't they use immune competent mice of the FVB/N background of their original transgenic mice?

---

## [Author Response]

Reviewer #1:General assessment:To establish a model that enables the metastasis of murine MYC-driven HCC, the authors introduced Twist1, which is known to promote the metastasis of cancer cells, to their previously reported murine MYC-driven HCC model (Tet-off conditional expression system).Here they present a mechanism by which MYC and Twist1 drive the metastasis of HCC. This study represents technically solid work. It is striking that Twist1 expression increases the probability of the metastasis of MYC-driven HCC by 45 folds.However, the conceptual novelty of the suggested mechanism is somewhat limited by a previous publication (Cancer Res. 2013. Twist1 induces CCL2 and recruits macrophages to promote angiogenesis) that reported the same role of Twist1 for recruiting macrophages through CCL2 and promoting angiogenesis around carcinoma cells. Given that angiogenesis is required for metastasis, this study confirms the same role of CCL2 in the metastasis of liver cancer while reporting a new role of IL13 in M2 polarization in this process. The authors should discuss this reference in more detail to highlight the novelty of this study.

We thank the reviewer for this comment regarding conceptual novelty and mechanistic insight. We explain now, in more detail, the novelty of our work and we have revised the text to reflect the thoughtful comments of the reviewer.

We would like to briefly highlight what we believe is the conceptual novelty of our work.

First, we identify that *MYC* and *Twist1* mediate a surprising non-cell intrinsic mechanism of cooperation through a novel autochthonous transgenic mouse model of metastatic HCC. We demonstrate that *MYC* and *Twist1* cooperate to induce metastasis. *MYC* has not been thought of as a metastasis gene. *Twist1* has been thought of as metastasis gene but through effects on EMT. We show that together they cooperate to drive a transcriptional program that elicits a change in the host immune system.

Second, we specifically identify that *MYC* and *Twist1* cooperate to induce a cytokinome which is both required and sufficient to induce HCC metastasis.

Third, we show that *MYC* and *Twist1* are part of a generalizable mechanism to human cancer by showing that it is combined overexpression of *MYC* and *Twist1* that in 33 different human cancers (10,000 cancer specimens from TCGA) is generally associated with poor prognosis, pro-metastatic cytokines and TAM infiltration.

Fourth, we have gained specific new insights into how *Twist1* through CCL2 and IL13 mediates metastasis. The reviewer is certainly very correct to note that *Twist1* has been shown to induce CCL2 in breast cancer cell lines and recruit macrophages. However, we are able to show that *Twist1* alone expression in the liver does not lead to immune activation or spontaneous tumorigenesis. *Twist1* only in the context of cooperation with the *MYC* promotes metastasis. Moreover, treatment of *MYC*-HCC with CCL2 alone led to macrophage recruitment to the tumor but it was not sufficient to induce metastasis in the absence of the TAM-polarizing cytokine IL13.

To address the prior relevant work, we have now expanded the Discussion to include a specific reference of the Cancer Res. 2013 manuscript and also explained the novelty of our manuscript in this context as follows:

“Low-Marchelli and colleagues previously reported that *Twist1* transcriptionally induces CCL2 in breast cancer cell lines which leads to macrophage recruitment (Low-Marchelli et al., 2013). Our study demonstrates that, in fact, CCL2 induced macrophage recruitment alone was not sufficient to cause metastasis in vivo and IL13 induced macrophage polarization plays an essential and complementary role in promoting angiogenesis and metastasis.”

An interesting twist of this study is that Twist1 somehow needs MYC to promote the metastasis of liver cancer. Given that MYC and Twist1 have been reported to cooperate, if the authors can shed some light on the mechanism underlying this dependency, it will increase the impact of this study. Does MYC increase the transcriptional factor activity of Twist1, for example?

The reviewer makes an outstanding point that our “twist” is that “*Twist1*” needs *MYC* and mechanistic insight on this dependency would be exciting. So, to address this mechanistic question, we now add two results. First, we examined if both *MYC* and *Twist1* need to be constitutively active to sustain metastasis; and second we examined if *MYC* and *Twist1* were directly coordinating to regulate cytokine expression we implicated in the mechanism of metastasis.

First, we provide evidence using additional new data that the mechanism really is that you need both transcription factors to be present simultaneously and constitutively proving this is an active cooperative and sustained mechanism, as opposed to *Twist1* or *MYC* enabling distinct programs during tumor progression. To do this, we generated new tumor derived cell lines whereby we could discriminate the independent roles of *MYC* and *Twist1* in metastasis by modulating the expression of *MYC* and *Twist1* separately. Primary tumor derived cell lines from *MYC/Twist1* HCC were retrovirally transduced with constitutive *MYC* and/or *Twist1*, such that upon inactivation of transgene expression with Doxycycline, they now constitutively expressed *MYC* and/or *Twist1* (new Figure 3D). We confirmed that treatment of these cell lines with doxycycline resulted in the continued expression of constitutive *MYC* and/or *Twist1* by qPCR (new Figure 3—figure supplement 1A). We observed that the inactivation of either *MYC* or *Twist1* abrogated the ability of the cells to develop lung metastasis when injected intravenously in NSG mice, while cells expressing both *MYC* and *Twist1* led to development of extensive lung metastasis with prominent macrophage infiltration (new Figure 3E-3F, Figure 3—figure supplement 1C). Thus *MYC* and *Twist1* cooperate, and are both required to be active constitutively and function together to induce metastasis. Our results collectively suggest the mechanism is an active transcriptional mechanism, that enables resulting HCC to elicit macrophage recruitment and polarization to drive metastasis.

We have included this new data in paragraph three of subsection “*MYC* and *Twist1* cooperate to remodel the tumor immune microenvironment”, new Figure 3 and new Figure 3—figure supplement 1.

Second, we examined whether *MYC* and *Twist1* as transcription factors, were both required to epigenetically regulate the cytokines, CCL2 and IL13. We identified *MYC* and *TWIST1* promoter binding upstream of human CCL2 and IL13 protein-coding genes in Gene Transcription Regulation Database (GTRD), a meta-analysis of ChIP-seq experiments (Dreos et al., 2017; Yevshin et al., 2017) (Figure 5—figure supplement 1E). Both *MYC* and *Twist1* demonstrated binding at multiple sites in the promoter region of CCL2 and IL13 in the ChIP-seq data from several different cancer cell lines (Figure 5—figure supplement 1E). We also looked for *MYC* and *Twist1* promoter binding sites in mouse CCL2 and IL13 promoters using motif finding analysis of the public data from JASPAR (Bryne et al., 2008) and Eukaryotic promoter database (EPD) (Dreos et al., 2017). Again, we found multiple potential *MYC* and *Twist1* transcription factor binding sites for both CCL2 and IL13 (Figure 5—figure supplement 1F). These data suggest that *MYC* and *Twist1* cooperate to transcriptionally regulate expression of CCL2 and IL13 in the cancer cells.

The new paragraph has been added in subsection “*MYC* and *Twist1* reprogram the crosstalk between cancer cells and macrophages” paragraph four. New figures have been added to Figure 5—figure supplement 1.

We also modified the Discussion section to reflect the results and added the following paragraph: “Our data suggests that *MYC* and *Twist1* cooperate to transcriptionally regulate the cytokinome landscape. We found multiple instances of *MYC* and *Twist1* promoter binding both in human and mouse CCL2 and IL13 genes by analyzing publicly available ChIP-seq data. *Twist1* is a basic helix loop helix (bHLH) transcription factor which has recently been shown to promote invasion of *MYCN* to enhancer sites thus potentiating its pro-proliferative function of *MYCN* in cancer (Zeid et al., 2018). We posit that a similar mechanism for epigenetic cooperation between *MYC* and *Twist1* likely exists in the context of hepatocellular carcinoma.”

Is IL13-mediated M2 polarization required for angiogenesis?

We thank the authors for raising this questions about angiogenesis. We have now added data showing that in vivo treatment of *MYC*-HCC with combination of CCL2 and IL13 promoted metastasis by stimulating angiogenesis. The most significant increase in angiogenesis, as assessed by CD31 IHC staining, was noted in *MYC*-HCC tumors in mice treated with CCL2 and IL13.

The new text has been added to subsection “Ccl2 and Il13 are sufficient and required for metastasis of HCC”, paragraph two and the data to Figure 6—figure supplement 1B-C.

Regardless of this gap of knowledge, this study will be of interest to the liver cancer community now that the authors show that the combined inhibition of CCL2 and IL13 suppress the metastasis in a mouse model of liver cancer and that the plasma level of IL13 correlates with the vascular invasion in HCC.

Thank you very much for your kind comments regarding the potential value of our work to the liver cancer community.

Reviewer #2:[…]1) The work shows that MYC induces the expression of CCL2 and IL13 and that these cytokines are sufficient for the MYC-induced liver tumors to metastasize. This is a relevant information. However, whether IL13 or CCL2 are direct transcriptional targets of MYC+TWIST1 remains to be elucidated. The authors may at least ask in the databases whether the DNA region around the transcription start site of IL13 and CCL2 genes contain MYC and Twist1 binding motifs and to show MYC and Twist1 binding by ChIP. Actually the binding of MYC to the genes can be also tested I many cell lines from the public data of the ENCODE consortium.

We thank the reviewers for raising this question. We have now done precisely what the reviewer suggested. In this study and examined if CCL2 and IL13 are direct transcriptional targets. Also see answer to reviewer #1 above.

*MYC* and *Twist1* are both transcription factors, so we evaluated if they epigenetically regulated CCL2 and IL13 expression. We identified *MYC* and *TWIST1* promoter binding upstream of human CCL2 and IL13 protein-coding genes in Gene Transcription Regulation Database (GTRD), a meta-analysis of ChIP-seq experiments (Dreos et al., 2017; Yevshin et al., 2017) (Figure 5—figure supplement 1). Both *MYC* and *Twist1* demonstrated binding at multiple sites in the promoter region of CCL2 and IL13 in the ChIP-seq data from several different cancer cell lines (Figure 5—figure supplement 1E). We also looked for *MYC* and *Twist1* promoter binding sites in mouse CCL2 and IL13 promoters using motif finding analysis of the public data from JASPAR (Bryne et al., 2008) and Eukaryotic promoter database (EPD) (Dreos et al., 2017). Again, we found multiple potential *MYC* and *Twist1* transcription factor binding sites for both CCL2 and IL13 (Figure 5—figure supplement 1F). These data suggest that *MYC* and *Twist1* cooperate to transcriptionally regulate expression of CCL2 and IL13 in the cancer cells. The new paragraph has been added in subsection “*MYC* and *Twist1* reprogram the crosstalk between cancer cells and macrophages” paragraph four. New figures have been added to Figure 5—figure supplement 1.

To discriminate the independent roles of *MYC* and *Twist1* in metastasis we developed cell lines where we could modulate the expression of *MYC* and *Twist1* separately. Primary cell lines from *MYC/Twist1* HCC were retrovirally transduced with *MYC* and/or *Twist1*, such that upon inactivation of transgene expression with Doxycycline, they now constitutively expressed *MYC* and/or *Twist1* (new Figure 3D). We confirmed that treatment of these cell lines with doxycycline resulted in the continued expression of constitutive *MYC* and/or *Twist1* by qPCR (Figure 3—figure supplement 1A). We observed that the inactivation of either *MYC* or *Twist1* abrogated the ability of the cells to develop lung metastasis when injected intravenously in NSG mice, while cells expressing both *MYC* and *Twist1* led to development of extensive lung metastasis with prominent macrophage infiltration (new Figure 3E-3F, Figure 3—figure supplement 1C). Thus *MYC* and *Twist1* cooperate, and are both required to induce metastasis of HCC by a macrophage dependent mechanism.

We have included this new data in paragraph three of subsection “*MYC* and *Twist1* cooperate to remodel the tumor immune microenvironment”, new Figure 3 and new Figure 3—figure supplement 1.

2) Figure 1C: It is difficult to tell from these images whether the luciferase is being expressed in the liver of the animal, or another organ of the body or in the peritoneum. Maybe they can use another technique to show Twist1 in the liver upon Dox withdrawal. For example and IHQ or IF for Twist or Luc, a western blot or even RT-PCR showing the Twist1 RNA expression).Also, they should give information on the penetrance of the MYC-Twist1/Luc. What percentage of mice show it? At the same intensity? It does not look so from the two mice of the figure.

Thank you very much for pointing out that the BLI images were not clear. We have now included new Figure 1Cwhere the luciferase activity in vivo, as assessed by bioluminescent imaging, is clearly localized to the liver.

We have now also included qPCR and immunoblotting data in new Figure 1—figure supplement 1B-Cshowing inducible *MYC* and *Twist1* expression in the tumors upon oncogene activation and inactivation.

Also, you asked us a question about the penetrance of the mouse model. The *MYC/Twist1*-HCC model has high penetrance. We find that all transgenic mice show Luc expression in the liver when imaged on BLI. Almost all mice we have examined developed liver tumors with high *MYC*+*TWIST1* (95%). Among the *MYC/Twist1* mice who develop liver tumors, >90% develop metastases.

3) Figure 1—figure supplement 1B: As the authors rightly say, it is important to rule out an effect of Twist1 on MYC levels. However, the figure apparently shows an increase in MYC levels in the Twist1-MYC livers, if the reduced levels of Actin are considered (compare lanes 1 and 3). A better western or a quantification of this blot normalizing against actin would be required to make the conclusion.

To address the reviewers concern regarding analysis of *MYC* levels, we have now quantified the immunoblot and we found no significant difference in *MYC* and *MYC/Twist1*-HCC. But we agree with the reviewers that the immunoblotting images needed refinement. We have now repeated this experiment using three *MYC*- and three *MYC/Twist1*-HCC and added new Figure 1—figure supplement 1C with quantification of blot using relative densitometry. We do not find any statistic difference in *MYC* levels between *MYC*- and *MYC/Twist1*-HCC. We have also shown by new data added in Figure 1—figure supplement 1B (qPCR) that confirm that *MYC* mRNA and protein levels are statistically similar between *MYC*- and *MYC/Twist1*-HCC

4) Subsection “MYC and Twist1 cooperate to remodel the tumor immune microenvironment” paragraph two, Figure 2—figure supplement 1E: "the inactivation of MYC and Twist1 in tumors shows rapid tumor regression". However, no quantification is given on the extent of tumor repression upon the addition of Dox to the water. In how many mice? Do the tumors completely regress? How long before they regress?

Thank you for asking us to clarify details regarding tumor regression. We have now included more details on tumor regression timeline. We see complete macroscopic tumor regression in all observed mice (n=20). Tumors begin regressing within a day of oncogene inactivation and completely regress within 2-3 weeks. These details have now been added to the text.

5) Figure 4C: It is known that the macrophage subtype M2a is the one activated by IL13. So it will be interesting to check whether this is the subtype polarized in the MYC+TWIST1 liver tumors.

The reviewer brings up an interesting point regarding macrophage polarity and subclasses. M2a is classically characterized by expression of CD206, and Arg1. We have shown that, in fact, CD206 and Arg1 are significantly overexpressed in the *MYC/Twist1*-HCC compared to *MYC*-HCC (Figure 2F). Also treatment of monocytes with supernatant from *MYC/Twist1*-HCC cells in vitroleads to polarization to M2a-like phenotype with higher expression of Cd206 and Arg1 as shown in Figure 5C.

To further characterize these macrophages, we now additionally evaluated expression of another M2a macrophage marker CD163 and found that *MYC/Twist1*-HCC had significant higher infiltration of CD163+ M2a-like macrophages compared to *MYC*-HCC. We have now added a sentence to the Discussion clarifying that polarization to M2a-like state is observed in *MYC/Twist1*- HCC.

Lastly, we have included new data demonstrating polarization to CD206+ M2a-like macrophages when mice bearing *MYC*-HCC are treated with IL13 either alone or in combination with CCL2+IL13 (Figure 6—figure supplement 1A and 1B)

*6) Subsection “Combined inhibition of CCL2 and IL13 synergistically inhibits metastasis* in vivo*”: The liver tumor burden in response to the neutralizing antibodies must be quantified (e.g. by densitometry of the images)*

Thank you, we appreciate that quantification of the liver tumor burden is important. We have now compared the primary liver tumor burden in the antibody treated mice by quantifying the volume of the primary orthotopic tumor (new Figure 7A) and also quantified bioluminescent imaging (new Figure 7B). We do not find a statistical difference in the tumor burden although there is a trend for improvement between control antibody treated mice and mice treated with anti CCL2+IL13 (p=0.07). The slight difference in BLI signal is likely reflecting the metastatic tumor burden since the size of primary tumor was clearly not different between the 4 groups.

Reviewer #3:General assessment:The manuscript by Baylot et al. shows a new metastastic mouse model of HCC, based on the cooperative function of MYC and Twist 1. The authors provide evidence that MYC and Twist1 together are responsible for the induction of a cytokinome, including CCL2 and IL13, which cause macrophage tumor infiltration and their transition to a metastasis-promoting M2 phenotype. The story is very interesting by itself, but it is probably just the beginning of further studies needed to define the distinct contributions of MYC and Twist1 to the metastatic phenotype (unfortunately MYC and Twist1 are both under the same Tet System control and cannot be turned off separately in the MYC/Twist1 mice).

The reviewer raises an important concern that it would be invaluable if we could independently regulate *MYC* and *Twist1*. To address this great idea, we have now included substantial additional data (also see responses to reviewers 1 and 2). We hope the reviewer will consider the additional new data, further confirmation of the required dependency of combined and continued expression of both *Twist1* and *MYC* to induce metastasis. To do this, we developed tumor derived cell lines where we could modulate the expression of *MYC* and *Twist1* separately. Primary cell lines from *MYC/Twist1* HCC were retrovirally transduced with constitutive *MYC* and/or *Twist1*, such that upon inactivation of transgene expression with Doxycycline, they now constitutively expressed *MYC* and/or *Twist1* (new Figure 3D). We confirmed that treatment of these cell lines with doxycycline resulted in the continued expression of constitutive *MYC* and/or *Twist1* by qPCR (Figure 3—figure supplement 1A). We observed that the inactivation of either *MYC* or *Twist1* abrogated the ability of the cells to develop lung metastasis when injected intravenously in NSG mice, while cells expressing both *MYC* and *Twist1* led to development of extensive lung metastasis with prominent macrophage infiltration (new Figure 3E-3F, Figure 3—figure supplement 1C). Thus *MYC* and *Twist1* cooperate, and are both required to induce metastasis of HCC by a macrophage dependent mechanism.

We have included this new data in paragraph three of subsection “*MYC* and *Twist1* cooperate to remodel the tumor immune microenvironment”, new Figure 3 and Figure 3—figure supplement 1.

Also, there is no mechanism explaining exactly how the macrophages are promoting metastasis, even though it is clear that they are needed for the process.

We thank the reviewers for raising this point about the mechanism by which macrophages promote metastasis in this model. We have added new data and demonstrate two mechanisms by which macrophages are promoting metastasis in *MYC/Twist1*-HCC (1) increasing invasiveness of cancer cells and (2) stimulating angiogenesis.

TAMs have been shown to increase the migratory capacity of tumor cells (Lin et al., 2001). To evaluate if this was the mechanism by which the M2-like TAMs in *MYC/Twist1*-HCC promote metastasis, we isolated TAMs from primary *MYC*-HCC and *MYC/Twist1*-HCC (Figure 3A). Conditioned media from TAMs isolated from *MYC/Twist1*-HCC but not *MYC*-HCC increased the invasiveness of both *MYC*- and *MYC/Twist1*-HCC tumor cells in vitro (Figure 3A-3C).

We have now added data showing that in vivo treatment of *MYC*-HCC with combination of CCL2 and IL13 promoted metastasis by stimulating angiogenesis. The most significant increase in angiogenesis, as assessed by CD31 IHC staining, was noted in tumors arising in mice treated with CCL2 and IL13 which showed recruitment and polarization to M2-like macrophages (new Figure 6—figure supplement 1C).

Summary of substantive concerns:1) General comment for figures: the scale bars are missing and sometimes the same figure includes the comparison among panels with different magnifications (i.e. Figure 1J-K, lung and liver panels, or Figure 2G-H). That needs to be corrected in all figures.

We thank the reviewers for pointing out the figures would be improved with scale bars. We have redone the figures to make sure the magnifications and scale bars are uniform throughout.

2) Regarding the same Figures 1J-K, in subsection “Twist1 induces spontaneous metastatic progression of MYC-driven HCC in vivo” the authors state that MYC levels are similar in the two tumor models, but the images do not seem to agree with that statement: in MYC/Twist1 HCC top left panel, MYC seems much higher that the corresponding MYC HCC panel, while in the merge it looks exactly the opposite. The authors should quantify the images in order to make conclusions and correct for different exposure times.

We apologize for the confusion regarding our data showing similar *MYC* levels. We have repeated these experiments, now included a new Figure 1J and K with clear demonstrations of equivalent *MYC* levels in both models using qPCR and western blotting.

We have also confirmed by immunohistochemistry that *MYC* levels are not different between *MYC*-HC and *MYC/Twist1*-HCC (not included in paper).

3) The authors state that "Macrophage depletion with clodronate liposomes […] had reduced intrahepatic and lung metastases". However, the figure clearly shows a difference also in the primary tumor burden, which could in part account for the reduction in mets. This is definitely a point that needs to be clarified.

We appreciate the reviewer’s concern that macrophage depletion of macrophages may be associated with reduced primary tumor burden. We quantified mouse liver tumor burden but did not find a statistical difference between control or clodronate treated mice primary tumors. We have now included quantification of BLI imaging from primary liver tumor burden in all mice. We have also increased the representative BLI imaging to show 3 mice in each group (Figure 4E). There was no statistically significant difference in densitometry between control treated and clodronate treated mice (p=0.603). We do not believe that the complete abrogation of metastasis upon treatment with liposomal clodronate can be explained by the slight decrease, albeit nonsignificant, in primary tumor growth. We have included new text in paragraph one of subsection “*MYC* and *Twist1* reprogram the crosstalk between cancer cells and macrophages” and the new data has been included in Figure 4E.

4) The authors should at least comment on the fact that Twist1 is a bHLH protein that can bind the same Ebox as MYC does, hence not only possibly cooperating, but even competing with it. Again, some characterization of this molecular aspects of target regulation might be relevant to future studies.

We appreciate this important suggestion that we should address possible epigenetic cooperation between *MYC* and *Twist1* as they can bind to same Eboxes. We have now added new data exploring the mechanism by which *MYC* and *Twist1* cooperate. Please also see our responses to reviewer #1 (point one) and reviewer #2 (point one).

Briefly, we evaluated epigenetic cooperation between *MYC* and *Twist1* in the regulation of CCL2 and IL13 expression. We identified *MYC* and *TWIST1* promoter binding upstream of human CCL2 and IL13 protein-coding genes in Gene Transcription Regulation Database (GTRD), a meta-analysis of ChIP-seq experiments (Dreos et al., 2017; Yevshin et al., 2017) (Figure 5—figure supplement 1E). Both *MYC* and *Twist1* demonstrated binding at multiple sites in the promoter region of CCL2 and IL13 in the ChIP-seq data from several different cancer cell lines (Figure 5—figure supplement 1E). We also looked for *MYC* and *Twist1* promoter binding sites in mouse CCL2 and IL13 promoters using motif finding analysis of the public data from JASPAR (Bryne et al., 2008) and Eukaryotic promoter database (EPD) (Dreos et al., 2017). Again, we found multiple potential *MYC* and *Twist1* transcription factor binding sites for both CCL2 and IL13 (Figure 5—figure supplement 1F). These data suggest that *MYC* and *Twist1* cooperate to transcriptionally regulate expression of CCL2 and IL13 in the cancer cells. The new paragraph has been added in subsection “*MYC* and *Twist1* reprogram the crosstalk between cancer cells and macrophages” paragraph four. New figures have been added to Figure 5—figure supplement 1.

Based on the reviewer's suggestion to discuss the E box binding by *MYC* and *Twist1*, we have also added this new paragraph to the Discussion-

“Our data suggests that *MYC* and *Twist1* cooperate to transcriptionally regulate the cytokinome landscape. We found multiple instances of *MYC* and *Twist1* promoter binding both in human and mouse CCL2 and IL13 genes by analyzing public data. *Twist1* is a basic helix loop helix (bHLH) transcription factor which has recently been shown to promote invasion of *MYCN* to enhancer sites thus potentiating its pro-proliferative function of *MYCN* in cancer (Zeid et al., 2018). We posit that a similar mechanism for epigenetic cooperation between *MYC* and *Twist1* likely exists in the context of hepatocellular carcinoma.”

5) Orthotopic and intravenous transplantation assays: To perform this in vivo assay the authors state that the tumors or cells are inoculated into NSG mice. Why do the authors perform these experiments with an immunodeficient mouse strain? Why don't they use immune competent mice of the FVB/N background of their original transgenic mice?

The reviewer raises a good question of why the orthotopic transplantation assays were performed in immunocompromised mice and not in WT FVB mice. As has been observed for many transgenic models, we were not able to transplant primary tumors into new syngeneic wild type FVB/N with regularity (10% success rate). We believe this is likely explained by possible allogeneic rejection of donor-specific antigens in the transplanted tumor by the host immune system. Hence, we were concerned that experiments would reflect a strong selection that could markedly bias the results. However, we think it is very important to point out that most of the experiments we performed were done in a primary transgenic mouse model and that this strength and novelty of our findings is exactly that we did experiments in primary tumors in vivo.

We point out that the existing syngeneic allograft orthotopic transplant models have used immortalized cell lines like MC38 colon cancer and B16 melanoma cell lines in C57/B6 background. Our cell lines derived from primary tumors are semi-immortalized and likely are still highly antigenic. Another possibility is that the macrophages recruited by the *MYC/Twist1*- HCC cells are able to directly and specifically reject allografts in a CD8^+^ T cell dependent manner as has been shown recently (Chu et al., 2019).

Although experiments in wildtype mice would be ideal, we hope that the experiments we have done in NSG mice are adequate for the purposes of this study. Our focus is on macrophage recruitment to the orthotopic liver tumor and metastatic sites. Macrophages are functional in NSG mice and have been used by several investigators for this purpose (Sukocheva et al., 2010; Behan et al., 2013) (Hu et al., 2011) (Krepler et al., 2004). Moreover, the main mechanisms by which we show that M2-like macrophages promote metastasis in the *MYC/Twist1* like stimulating angiogenesis and increasing cancer cell invasiveness are largely independent of adaptive immunity.

Finally, most of our experiments were performed in primary tumors in a new transgenic models of *MYC/Twist1* HCC where we could study the natural history of stepwise progression of metastatic HCC in an immunocompetent host.